# Provisioning of Fog Computing over Named-Data Networking in Dynamic Wireless Mesh Systems

**DOI:** 10.3390/s24041120

**Published:** 2024-02-08

**Authors:** Roman Glazkov, Dmitri Moltchanov, Srikathyayani Srikanteswara, Andrey Samuylov, Gabriel Arrobo, Yi Zhang, Hao Feng, Nageen Himayat, Marcin Spoczynski, Yevgeni Koucheryavy

**Affiliations:** 1Faculty of Information Technology and Communication Sciences, Tampere University, 33720 Tampere, Finland; dmitri.moltchanov@tuni.fi (D.M.); andrey.samuylov@tuni.fi (A.S.); yevgeni.koucheryavy@tuni.fi (Y.K.); 2Intel Labs, Portland, OR 97124, USA; srikathyayani.srikanteswara@intel.com (S.S.); gabriel.arrobo@intel.com (G.A.); yi1.zhang@intel.com (Y.Z.); hao.feng@intel.com (H.F.); nageen.himayat@intel.com (N.H.); marcin.spoczynski@intel.com (M.S.)

**Keywords:** fog computing, NDN, wireless mesh networks, service discovery, dynamic face management

## Abstract

Fog computing is today considered a promising candidate to improve the user experience in dynamic on-demand computing services. However, its ubiquitous application would require support for this service in wireless multi-hop mesh systems, where the use of conventional IP-based solutions is challenging. As a complementary solution, in this paper, we consider a Named-Data Networking (NDN) approach to enable fog computing services in autonomous dynamic mesh formations. In particular, we jointly implement two critical mechanisms required to extend the NDN-based fog computing architecture to wireless mesh systems. These are (i) dynamic face management systems and (ii) a learning-based route discovery strategy. The former makes it possible to solve NDN issues related to an inability to operate over a broadcast medium. Also, it improves the data-link layer reliability by enabling unicast communications between mesh nodes. The learning-based forwarding strategy, on the other hand, efficiently reduces the amount of overhead needed to find routes in the dynamically changing mesh networks. Our numerical results show that, for static wireless meshes, our proposal makes it possible to fully benefit from the computing resources sporadically available up to several hops away from the consumer. Additionally, we investigate the impacts of various traffic types and NDN caching capabilities, revealing that the latter result in much better system performance while the popularity of the compute service contributes to additional performance gains.

## 1. Introduction

Fog computing is an emerging data-processing concept poised to significantly influence future networks and Internet of Things (IoT) applications [1,2]. It notably facilitates data processing at end-user devices, as opposed to relying solely on dedicated edge or cloud servers [3]. The recent unprecedented proliferation of wireless connectivity, encompassing direct device-to-device (D2D) communications technologies supported by the IEEE (WiFi-direct [4]), and cellular standards (LTE- and NR-sidelink [5,6,7]) raises the question of how to efficiently extend fog computing architecture to exceptionally dynamic wireless systems, like mobile mesh networks.

Conventionally, fog computing relies on an IP-based architecture, such as in [1,8,9], and requires the domain name resolution system (DNS) for resource discovery. Although several approaches are enabling this functionality in wireless mesh systems [10,11], the performance of these is still far from optimal due to the intermittent nature of wireless links.

Named-Data Networking (NDN) offers new opportunities for enhancing user experiences in scenarios where traditional IP networking is challenged [12,13,14,15]. NDN eliminates the necessity of maintaining communication sessions between explicitly addressable hosts and shifts the focus to the requested content instead [16]. This is achieved by utilizing an ’Interest’ packet to request specific data, irrespective of its location. When a node possesses the requested, it responds to the Interest packet with the data. Notably, the NDN architecture is also inherently agnostic concerning the type of the requested content, rendering it a fitting solution for fog computing applications [17,18].

Fog computing in wireless mesh networks represents an exceptionally dynamic system characterized by its heterogeneous resources. The core factors contributing to this dynamism include node mobility, continuously changing wireless link states, fluctuating availability of computing resources at nodes, and the variability in user requests. Resources, including computing resources, storage, raw sensor data, and functions, are distributed among various proximate end devices. The most efficient way to address such dynamism is to harness all locally available resources and enable multi-tenancy for the resources and computations of the nodes with inherently different capabilities.

The authors of [19] propose a framework encompassing heterogeneous IoT resources in a given region and variations in processing and communication offloading that could be leveraged by the fog IoT architecture. The architecture relies on an IoT-in-the-fog controller that can probe local resources and communicate directly with a local fog mediator, which could be the cloudlet/edge access point. In [20], the authors considered an inter-NDN MANET scenario and overcame session disruptions when crossing MANETs, which ran over different wireless technologies and protocols. The authors of [21] extended the conventional fog-to-cloud communication paradigm by introducing a horizontal fog-to-fog communication layer. This layer utilizes information-centric networking (ICN) to lessen cloud dependency and enhances the fog layer with name-oriented communications, in-network caching, and built-in mobility support to achieve lower latency, better mobility, and higher data communication efficacy for fog computing. Furthermore, to take full advantage of ICN, a fog caching scheme has been enhanced with the in-network caching capability of ICN end-user devices to facilitate content delivery in the IoT environment [22]. The author of [23] developed user-centric and delegated mode-based solutions for service provisioning in NDN systems, where the latter approach presumed outsourcing the service search functionality to the network node. An automated in-network service construction over NDN was provided in [24]. Finally, [25] introduces advancements in service discovery within NDN, enabling exhaustive exploration of network segments.

One of the principal challenges in advancing fog computing over NDN is the current lack of support for multi-hop wireless networks. As of now, a single NDN Forwarding Daemon (NFD) face is employed for communication with all the neighboring devices in wireless setups. This face is bounded by a specific wireless interface (e.g., WiFi board/network interface card (NIC)) configured in broadcast mode. Given that the default NDN technology imposes restrictions on sending a packet via the face from where it was received, wireless communication in NDN is limited to a single hop. This issue also raises problems related to communication reliability in wireless environments, as broadcast wireless communications typically do not support retransmissions. Furthermore, it is important to highlight that the forwarding strategies proposed for NDN have been designed with wired networks in mind, where changes in network topology are infrequent.

The potential applications of Named-Data Networking (NDN) and fog computing span diverse scenarios, each demonstrating the strengths of this technology in dynamic and resource-constrained environments:Mobile network nodes in smart cities: A quintessential example is the smart city infrastructure, where mobile nodes such as vehicles or mobile devices continuously move, dynamically altering the network topology. Leveraging NDN with fog computing in such environments can facilitate efficient data dissemination and retrieval. This combination is particularly beneficial in reducing latency and enhancing data availability at the network’s edge, crucial for real-time applications like traffic management and event streaming.Drones in Mesh Networks: Another promising application involves the use of drones in mesh networks for agricultural monitoring, disaster management, or delivery services. Integrating Fog Computing with NDN here enables drones to share vital information in a robust and decentralized manner. The information could be, for example, weather data or emergency signals. This approach is especially beneficial in areas with limited infrastructure, as it supports autonomous and resilient drone operations.

Further enhancing these scenarios is the application of a multi-stage machine-learning (ML) overlay, which brings capabilities like predictive analytics and adaptive routing. This advancement improves the efficiency and adaptability of the systems, significantly elevating the overall performance and user experience.

This paper addresses the identified gaps and refines existing solutions by proposing two mechanisms that enable efficient NDN-based fog computing services in wireless mesh systems: (i) dynamic face management capability and (ii) adaptive, learning-based forwarding strategy. The first mechanism enhances efficient communications by utilizing unicast transmission mechanisms, such as request-based medium access (RTS/CTS) and acknowledging packets successfully received in WiFi. The second, an adaptive learning-based forwarding strategy, efficiently reduces network interference by forwarding Interest packets towards their intended destinations. Importantly, these two functionalities maintain compatibility with existing NDN stacks, ensuring full backward compatibility with current NDN implementations. Having implemented this system in ndnSIM, an extension module for NS-3 (a network-level simulator, NLS), we subsequently evaluate and characterize the performance of the proposed solution.

The main contributions of our study can be summarized as:joint implementation and evaluation of the unicast Ethernet method faces in NFD, enabling reduced overheads (as compared to broadcast) when forwarding Interest packets in dynamic mesh networks;performance evaluation of the adaptive learning strategy and dynamic Face management system in dynamic network conditions showing that the suggested enhancements in NDN system design efficiently support fog computing in multi-hop wireless mesh systems.

The paper is organized as follows. Section 2 overviews the prior art. Section 3 elaborates on our proposed solutions, including the dynamic Face management system (Section 3.1) and adaptive forwarding strategy (Section 3.2). The performance evaluation of the proposed enhancements is discussed in Section 4. Conclusions are drawn in the last section.

## 2. Background and Related Work

Given the inherently unreliable nature of broadcast wireless transmission mediums and the challenge posed by NDN’s inability to operate over such inherently broadcast mediums, these emerge as the two principal components limiting NDN performance in multi-hop wireless mesh systems. This section presents a review of the related work addressing these functionalities. Additionally, we discuss the forwarding strategies that have been proposed to date.

### 2.1. NDN in a Multi-Hop Wireless Environment

#### 2.1.1. Problem Statement

The Named-Data Networking (NDN) protocol uses the Named-Data Networking Forwarding Daemon (NFD) to determine the destination of an Interest packet [26]. This includes a set of tools for routing and forwarding Interests and for sending data packets back along the same path discovered by the Interest packets. When the NFD selects an outgoing Face for an Interest packet, it performs a lookup on the Forwarding Information Base (FIB). In contrast to IP networks, NDN does not support node IDs. Instead, forwarding relies exclusively on the node’s interface names (Face ID) to specify a next hop. Configuration of Faces includes remoteUri representing the remote endpoint, such as the MAC address of the remote device.

In NDN, the Face can function as a physical network interface for communication over a physical link, an overlay communication channel, or an inter-process communication channel between NFD and a local application [26]. In the context of wired communications (p2p link type, as per the NFD terminology), every connection to a neighboring node utilizes a distinct Face, as depicted in Figure 1. This approach allows the NFD to determine the path for an interest packet uniquely and prevent flooding. In wired networks, each Face of a node is linked to only one neighboring device, facilitating this specificity.

The Named-Data Networking Forwarding Daemon (NFD) is a key element in the architecture of Named-Data Networking (NDN). The incorporation of UnicastEthernetTransport has markedly improved NFD’s ability to manage faces. UnicastEthernetTransport enables NFD to create a separate unicast face for each peer on both wired and wireless Ethernet adapters. UnicastEthernetTransport in NFD is designed to handle unicast communications over Ethernet. It supports the creation of individual unicast faces for each peer, which is crucial for targeted communication in NDN. This transport mechanism allows for efficient management of network interfaces, binding each unicast face to a specific network interface. This is particularly beneficial in environments with multiple interfaces, such as mesh networks. It supports both unicast and multicast modes of communication, making it versatile for various networking scenarios. Unicast mode is particularly useful for direct, point-to-point communication, while multicast mode facilitates the distribution of data to multiple recipients. One of the key features of UnicastEthernetTransport is its adaptability to both wired and wireless Ethernet adapters, treating them uniformly. This enhances the flexibility of NFD in different network setups. The integration of UnicastEthernetTransport significantly extends the capabilities of NFD, particularly in handling complex network structures and traffic patterns more efficiently. UnicastEthernetTransport plays a vital role in enhancing the capabilities of NFD within NDN. Its dynamic management of multiple faces and support for both unicast and multicast communications across varied interfaces cements NFD’s position as a robust and adaptable network data management tool. [27]

#### 2.1.2. Related Work

Initial implementations of multi-connectivity scenarios, such as wireless mesh networks, rely upon the broadcast communication mode [28]. The broadcast-based networking causes a broadcast storm in multi-hop networks [29]. Although the broadcast storm in NDN is less severe than in TCP/IP networks, due to the default loop detection mechanism implemented in NFD, the negative effect on network performance is still significant. As detailed in [30], the high collision rate at the MAC layer is further amplified with an increased number of consumers and their transmission rates. These collisions lead to higher packet loss and latency, see Figure 2.

Several solutions have been proposed to handle the broadcast storm problem in wireless networks at higher layers. In [31], an NDN-based vehicular network is introduced. Although this solution still employs broadcast mode, the authors devise a mechanism to limit Interest propagation in the network. This is achieved using the GPS coordinates of the nodes and a specially designed namespace. Furthermore, Ref. [30] presents a solution that restricts the forwarding of Interest packets to a limited number of hops. Although these solutions mitigate network overheads and collisions to some extent, their implementation is confined to particular scenarios. For example, solutions based on external geographical information (such as GPS coordinates) are not suitable for scenarios where such information is unavailable, like indoor environments. Moreover, due to the use of low-order modulation and coding schemes in the broadcast regime by some wireless technologies, the provided channel capacity is lower than the one provided in a unicast communication mode. Paper [32] explores the integration of integrating Software-Defined Networking (SDN) with Named-Data Networking (NDN) in wireless mesh networks. The study explores the potential of SDN in enhancing the efficiency and scalability of NDN deployments. It provides experimental evidence demonstrating how SDN can dynamically manage network resources, optimize data flow, and improve overall network performance in wireless mesh environments. This research is pivotal as it addresses the challenges of deploying NDN in wireless contexts, offering solutions that could be crucial for the future of wireless network architectures. Work [33] investigates how different forwarding strategies impact the traffic load in Named-Data Networks. Utilizing Mini NDN for simulation purposes, the authors offer a comprehensive analysis of various forwarding mechanisms. Their findings are significant for understanding and optimizing traffic management in NDN environments. The study’s insights into the efficacy of forwarding strategies provide a foundation for developing more efficient and reliable NDN systems, especially crucial as the network traffic continues to grow exponentially.

In [34], the authors proposed a timer-based solution to mitigate broadcast flooding in ICN-based multi-hop wireless networks. Upon the reception of the Interest or Data packet, the node randomly sets a timer for the received packet. The node rebroadcasts the packet when the timer expires. If this node hears the packet rebroadcasted by its neighbor node before the timer expires, it shall discard the packet immediately. In [35], the authors use a similar concept to reduce the Interest flooding in NDN-based vehicular networks. The timer calculation relies on both content connectivity score (Interest satisfaction ratio) and location score (if the current node is closer to the destination than the previous node). The node with a higher score can obtain a shorter timer to rebroadcast the Interest packet. Although the timer-based solutions can help to reduce the Interest/Data packet flooding, they still have the problem of uncertainty on the success of packet forwarding as there is no feedback.

In recent developments within the paradigm of Named-Data Networking (NDN), a significant focus has been on enhancing the efficiency of forwarding mechanisms. J. Shi et al. [36] pioneered a broadcast-based self-learning approach in NDN, presenting a paradigm where nodes autonomously learn the most efficient paths for data packet routing. This work laid the groundwork for further advancements in self-learning strategies within NDN, especially relevant for environments with dynamic topologies. Expanding on these ideas, T. Liang et al. [37] explored the application of NDN forwarding mechanisms in edge networks. Their study addressed the challenges of deploying NDN forwarders in scenarios where configuration and maintenance pose significant barriers. By proposing an “out-of-the-box” solution for NDN forwarders, they made strides towards simplifying the deployment of NDN in diverse network environments, including wireless mesh systems.

A significant portion of research in Information-Centric Networking (ICN) advocates for an application layer implementation as an overlay on the standard TCP/IP transport [38,39]. Using TCP/IP enables unicast communication in mesh networks and similar setups by default. However, the implementation of ICN as an overlay over the TCP/IP brings notable processing overheads. In the network’s relay nodes, each packet undergoes a process of decapsulation and is then directed to the application layer. This procedure results in extra latency and can adversely affect the energy efficiency of the system.

The authors in [40] consider a novel perspective on cache management in ICN. Baugh J. et al. introduce a per-face popularity approach to cache robustness, aiming to reduce cache pollution and improve hit rates. Their method represents a significant advancement in optimizing cache utilization in ICN within frameworks.

Liu et al. in [41] delve into the application of Named-Data Networking (NDN) in satellite communication. They propose a coded caching strategy specifically for satellite networks, addressing unique challenges like limited bandwidth and high latency. Their approach signifies a crucial step in enhancing data dissemination efficiency in space-based networks.

The Dynamic Unicast (DU) approach [42,43] has been proposed for Content-Centric Networking (CCNx), which is a framework for Information-Centric Networking. DU uses broadcast communication only until a content source has been found (single-hop discovery) and then retrieves content directly via unicast from the same source. However, even for the discovery phase, unicast communication can be more beneficial, enabling more sophisticated learning-based approaches in the forwarding plane and facilitating multi-hop resource discover.

### 2.2. NDN Forwarding Strategies

#### 2.2.1. Problem Description

Today, mesh networks, both stationary and mobile, mostly rely on reactive broadcast-based mechanisms to find packet forwarding paths. According to these mechanisms, every packet is transmitted to every neighboring node, facilitating the discovery of a path to the destination without the need to maintain routing tables and update them regularly. The benefits of reactive routing can be visible in mobile ad hoc and low-power wireless networks, where proactive routing protocols cause a lot of periodic routing announcements. However, the utilization of broadcasts dramatically increases network load.

The broadcast-based mechanism is well-suited for NDN architecture, as it does not depend on prior knowledge of the requested data location: (i) a consumer issues an Interest, (ii) the network finds a fitting producer to serve this Interest, (iii) and then the data are sent back in reverse direction to the originating node. With this idea in mind, efficient broadcast-based solutions for NDN systems should satisfy the following principles: (i) no global knowledge is available to any of the participating nodes, and each node makes an independent decision when and where to forward an incoming packet, (ii) no forwarded packets are altered in any way and no additional packets are generated, (iii) minimize the overheads of the broadcast by utilizing any prior possible knowledge gained through handling previous packets, and (iv) adaptation to changing environments caused by either increasing networks loads or mobility.

#### 2.2.2. Related Work

In the field of Information-Centric Networking (ICN), IP-based models significantly influenced the development of routing protocols, such as the Named-data Link State Routing protocol (NLSR) referenced by Wang et al. (2018) [44]. NLSR operates on two primary Link State Advertisements (LSAs): Adjacency LSA for broadcasting live link connections between an ICN/NDN router and its neighboring nodes and Prefix LSA for announcing registered name prefixes. To maintain network integrity, routers periodically exchange Link State Database (LSDB) hashes, facilitating the identification and rectification of any inconsistencies. This strategy of hop-by-hop synchronization effectively reduces unnecessary network traffic. Utilizing the LSAs received, NLSR generates a prioritized list of forwarding choices for each name prefix, thus supporting NDN’s flexible forwarding strategies. However, NLSR’s method of spreading neighbor information to all nodes can introduce significant routing overhead, particularly problematic in environments with dynamic node mobility and frequent topology changes, as NLSR was originally designed for more stable, wired networks.

The Content Connectivity and Location-Aware Forwarding (CCLF) protocol, designed for vehicular ad hoc networks (VANETs) within the Named-Data Networking (NDN) framework, offers an innovative approach to network routing [35]. In CCLF, vehicles act as intermediate nodes, employing broadcast transmission for forwarding. The protocol determines the forwarding path by calculating scores based on the vehicle’s current location, the previous hop’s location, the destination’s location, and the ratio of satisfied content requests. This method assumes knowledge of the destination’s location, a challenging requirement in edge networks where precise locations of computing nodes or data producers may be unknown.

Broadcast-based self-learning forwarding, a method used in both wired and wireless networks, represents a significant advancement in NDN. This technique involves broadcasting the initial packet along an unknown path and subsequently establishing forwarding routes based on the responses. In NDN environments, the Named-data Forwarding Daemon (NFD) uses the longest prefix match against the Forwarding Information Base (FIB) to decide on forwarding paths. This decision-making process may also incorporate criteria like energy efficiency [45] or load balancing [14]. Self-learning in NFD involves flooding the network with Interests when no FIB entry matches and learning from the paths of returned Data packets to create new FIB entries. Despite being advantageous, this method can cause network congestion due to the initial flooding necessary for path discovery.

E-CHANET, as discussed by Amadeo et al. (2013) [34], extends the AODV routing protocol for ICN in wireless networks. This protocol uniquely broadcasts both Interest and Data packets, an approach well-suited to the dynamic nature of wireless environments and the mobility of nodes. The broadcasting aspect of E-CHANET eliminates the need to continuously identify and update information about neighboring nodes. Additionally, it employs a timer-based forwarding strategy to control the potential issue of a broadcast storm effectively. Despite these advancements, E-CHANET faces challenges in practical wireless multi-hop network applications, particularly due to the high interference caused by unscheduled simultaneous transmissions and the lack of acknowledgment mechanisms, impacting its effectiveness.

## 3. The Proposed Solutions

In this section, we propose solutions to enable dynamic computing services over NDN. We start with the face management system, allowing for dynamic creation and removal of faces. The system enables unicast communications between nodes in the mesh wireless network, resulting in improved reliability on the data-link layer. Then, we propose a novel adaptive learning-based forwarding strategy.

### 3.1. Dynamic Face Management

#### 3.1.1. The Conceptual Approach

The workflow of the proposed solution, depicted in Figure 3, relies on continuous listening of the events on the channel. In particular, (1) when a wireless node comes into the coverage range of another one, the nodes trigger the Face Manager. Next, the Face Manager (2) creates a new Face for communication with the new neighbor node and (3) checks if the physical address (MAC address) of the peering node matches a record in the QDB. If the MAC address (remoteUri) of the peering node matches a record in the QDB, it means that these nodes have communicated recently, and related forwarding information is temporarily stored in the QDB after the communication channel was dropped. In this case, related forwarding information (4) migrates back from the QDB to the FIB while the matched record is (5) removed from the QDB. If (6) a link with one of the neighbor nodes drops, Face Manager deletes the Face associated with the dropped link. However, (7) Face configuration, including its remoteUri has to be stored in QDB for later use. Once done, FIB and PIT entries associated with the Face are also deleted, and the quarantine timer starts for the record created in the QDB. Once the timer expires, the entry is deleted from the QDB.

This dynamic Face management system enables wireless devices to utilize unicast mode, fully exploiting the benefits of advanced medium access mechanisms (such as RTS/CTS in WiFi) instead of using random access channel variations to transmit user data. Moreover, in unicast mode, wireless technologies support error-control methods, such as automatic repeat requests (ARQ), further improving communication reliability. Eventually, the dynamic Face management system promises a better user experience for customers in wireless mesh networking or other multi-connectivity scenarios. The next section provides an implementation example of the proposed solution for the case of WiFi.

#### 3.1.2. Implementation Details

In Named-Data Networking (NDN), the concept of a Face (implemented as the nfd::Face class) is instrumental in facilitating best-effort delivery services across various underlying communication mechanisms, such as sockets. This abstraction is key in abstracting the complexities and specific details of the underlying protocols, ensuring seamless integration with the forwarding layer of the NDN network [26]. Faces include two main parts—a link service and a transport service (NetDeviceTransport). Faces are created through callbacks by going through each network interface associated with the node (called NetDevices, ND), such as Ethernet port, WiFi board, etc. Configuration of the transport service includes a remoteUri attribute representing the remote endpoint of communication (for example, the MAC address of the neighbor node). This attribute is read-only (immutable during the Face lifetime). The process begins with identifying the scheme corresponding to the underlying protocol, such as Ethernet, in channel-level applications. This is followed by a representation specific to the scheme, which details the underlying address. An example of such an address in the context of Ethernet is *01:00:5e:00:17:aa* [26].

In our manuscript, we primarily focus on the application of our modifications within the ndnSIM environment, utilizing NetDevice and WifiNetDevice. However, it is important to note that in a practical, real-world deployment, the NFD interacts with Ethernet adapters through the libpcap library. Although the current scope of our study is centered around simulation, we acknowledge the potential and feasibility of adapting our proposed modifications for real-world implementation, see Figure 4.

### 3.2. Adaptive Forwarding Strategy

The NDN Forwarding Daemon supports modular design and extensibility for NDN protocol. At the data plane, its main functionalities are related to switching packets between its internal components. When the Forwarder receives a packet from an underlying link-layer service, it will consult a forwarding strategy for handling the packet, i.e., inserting, deleting, or updating any needed information in supplementary tables. Thus, by implementing a forwarding strategy, it is possible to control packet flows by making decisions on whether, when, and where to forward the packets.

To evaluate fog computing in dynamic wireless meshes, we considered a self-learning strategy considered in [37]. In our setting, this strategy attempts to find and maintain all (or at least a set of) possible routes by not only forwarding an interest through the already known route but also by probing other faces. This approach is also known as the probing mechanism in the Adaptive Smoothed Face-routing (ASF) strategy. The combined strategy leverages the benefits of the dynamic Face management system. The high-level algorithm behind the proposed combined strategy is illustrated in Figure 5 and consists of the following steps:*Initial broadcast search.* The incoming Interest packet is forwarded over all the outgoing interfaces when there is no knowledge available locally about the requested Name. Upon receiving such an Interest packet, a corresponding routing information entry is created.*Receiving data reply.* Receiving the Data packet over a certain Face implies that the producer is available via that Face. Thus, it can be further utilized for forwarding further Interest packets with the same Name.*Unicast search.* For a predefined time window, any other similar Names are forwarded directly through the explored direction (Face). During this time, all other Faces are not used. This allows the strategy to minimize broadcasting overhead.*Periodic probing.* Once in a predefined time window, after a new Interest packet with the already known Name is received, this Interest is sent over all the outgoing Faces that previously did not yield a data reply. This step is needed for both counteracting any possible network topology changes and discovery of other producers.

The algorithm of the considered strategy can be presented in two separate pipelines: Figure 6 shows inbound Interest packet processing and Figure 7 inbound Data packet processing. In both cases, the information about the packet passes from the Forwarder to the Strategy to decide how to proceed with this. See also Table 1.

The Interest-handling algorithm is illustrated in Figure 6. Upon receiving an Interest packet, the Self-Learning Strategy utilizes current local knowledge to decide which Faces to use for forwarding. If an Interest packet with a new prefix arrives, the Self-Learning Strategy creates a new entry for all available Faces. For all the previously explored Names, the Self-Learning Strategy holds a decaying time window. If a Face is marked positively, it means that a Data packet with the same Name has been handled before. Thus, this Face has the highest chance of receiving a reply again, and the Self-Learning Strategy forwards a copy of this Interest packet via this Face. At the same time, the strategy checks if the Face was marked positively for a certain period of time (the default value is 10 s, but it can be adjusted depending on the network dynamics) without receiving any Data replies. If this is the case, the strategy marks it as negative and sets a timer *FValue*.

When a Face is marked negatively, it is no longer used for forwarding purposes, while the values of *FValue* timer decreases with each time step. When the *CurFValue* timer reaches zero, and a copy of Interest is forwarded via this Face for probing. If the probing is not successful, i.e., no Data packet is received over this Face, the negative state of this Face is restored.

When the Self-Learning Strategy has to handle a Data packet, it follows the algorithm presented in Figure 7. First, the strategy sends the packet through the available back propagation Face(s) specified in the corresponding PIT entry. Then, it marks the inbound Face positively for the corresponding Name. This Face will then be used in the Interest handling pipeline to forward other Interest packets matching this Name.

An example of the forwarding time window dynamics is shown in Figure 8. For a new Name, all Faces are initialized with a predefined time window value, *FValueMaxBase*. This puts each Face into the “Exploit” regime, enabling broadcast search over all available Faces. To avoid Faces’ “saturation” when a chosen Face is utilized for a very long period of time, we limit the forwarding time window using *FValueMax*. The closer the current forwarding value *CurFValue* to its maximum, specified by *FValueMax*, the smaller the increment due to the successfully received Data packet. The probing time windows for a Face are represented by orange lines in Figure 8. When probing, the period between attempts increases with every unsuccessful attempt until *CurFValue* reaches the minimum value, specified by *FValueMin*.

With the proposed approach, the Self-Learning Strategy can handle any Interest and Data packets without global knowledge about the network topology, additional internode data exchange, or application configurations. At the same time, the decaying time window utilized by the Self-Learning Strategy can both minimize discovery overhead and handle situations such as link failures or producer re-discovery. The scalability of the proposed solution is ensured by controlling the frequency of flooding. The proposed strategy can also be further optimized to match the requirements of specific scenarios using the following extensions: (i) smart name aggregation into more generic prefixes over time to limit the required memory to store local routing information, (ii) decaying time window size adaptation in real time to account for rapid changes in topology for networking scenarios involving, e.g., nodes mobility, (iii) similarly to [46], “fish-eye” flooding can be implemented to limit the number of hops being flooded upon initial discovery.

We specifically note that the proposed Self-Learning Strategy is conceptually similar to the self-learning strategy in [37]. However, the proposed strategy attempts to find and maintain all (or at least a set of) possible routes by not only forwarding an interest through the already known route but also by probing other faces. The probe is limited by specific exponential timers attached to each face to limit the network flooding—the fewer successful data receptions one obtains through a given face, the less often one will probe it, while successful reception will increase the lifetime of the route.

## 4. Performance Evaluation

In this section, we analyze the performance of the NDN-based fog computing applications by utilizing network-level simulations (NLS) implemented in the NDN community simulator (ndnSIM) [47]. In particular, we start by describing the simulation environment. Then, we proceed by introducing the considered scenarios and data processing. Finally, we report our numerical results. We note that the baseline results for the system without the proposed enhancements are provided in [23]. We will refer to them throughout this section when discussing the presented results.

### 4.1. Simulation Setup

To produce our results, we upgraded ndnSIM with the compute service framework introduced in the previous work [23]. On top of that, we implemented in ndnSIM modifications described in Section 3. The main simulation parameters are summarized in Table 2. (There is something wrong with this sentence: Marcin) We specifically node that for static mesh environment, we position nodes on a grid separated by internode distance, *L*. The considered region is thus a square with an area given by $N{\left(L-1\right)}^{2}$ m^2^, where *N* is the number of nodes. In a mobile mesh environment, network nodes are initially positioned on a grid and then start to move according to a random direction model (RDM, [48]).

The simulator was configured as follows:**Simulation environment.** The considered scenario assumes mobile wireless nodes deployed in an area of limited size (free space environment).**Wireless channel.** Connectivity between nodes is enabled by IEEE 802.11n (5 GHz) wireless technology, with a range propagation loss model with the cutoff at 100 m and a Minstrel-HT rate adaptation algorithm.**Mobility model.** At the beginning of the simulation, wireless nodes were organized in a grid with a distance between nodes of 90 m. After the simulation started, wireless nodes started moving following the Random Direction Mobility (RDM) model.**Computing application.** In this simulation campaign, we assumed that there is a limited number of computing services available in the network. This assumption is motivated by a common IoT operation where devices (e.g., wearables) react to an event (e.g., weather notifications, traffic information) by utilizing standard processing algorithms (different types of software). This assumption justifies the overlapping of services requested by mobile users, supposing that users may request the same processing operations over the same data. More specifically, we considered 100 different software types that can be used to process data. Recognizing the varying popularity levels among software options, we extended our study beyond the uniform choice model, typically referred to as the Constant Bit Rate (CBR) traffic model, to incorporate the Zipf-Mandelbrot law for content popularity assessment. Furthermore, we imposed a constraint on the timeliness of computing results, setting their freshness threshold at one second. This approach implies that any data-processing outcomes are deemed obsolete after one second and are subsequently purged from the cache, provided caching is active. Detailed insights into the computing service methodology utilized in this analysis are elaborated in Pirmagomedov (2020) [23].

### 4.2. Data Processing and Metrics of Interest

To collect simulation data, we employed the method of replications [49]. For each set of input parameters, we run the simulator for 600 s of the system time one hundred times with different random number seeds. The results obtained in each run have been used to form a sample with independent identically distributed (iid) observations [50] and, thus, the results, presented in further subsections, are considered to be averaged over all the different seeds we have utilized. Given the extensive number of observations, the confidence intervals obtained are very narrow. Therefore, in the presentation of results, only point estimates are depicted.

We consider several metrics of interest, including the fraction of satisfied Interest requests and the mean full delay. In addition, we also report the share of received compute results in time. The latter metric integrates both latency and reliability in a single bundle, providing comprehensive insights into the system behavior.

### 4.3. Numerical Results

#### 4.3.1. Static Mesh Environment

We start investigating the system performance by reporting the fraction of satisfied Interest requests and mean full delay in Figure 9 for static mesh network conditions as a function of the frequency of Interest per second, 5×5 nodes configuration, enabled cache, multiple consumers, and single producer. Please note that from now on, the latter three parameters are encoded as a triple (C/N,K,M), where C/N defines whether cache enabled (*C*) or not (*N*), *K* denotes the number of consumers, *M* refers to the number of producers.

Analyzing the reported results, one may observe that for all the reported Interest frequencies, the mesh is capable of successfully satisfying the compute requests with the fraction reaching 1 for all considered numbers of consumers. The major differences between the considered cases lie in the mean full delay behavior reported in Figure 9b. Here, one may observe that the delay increases with the increased number intensity of Interests as well as with the increased number of consumers. However, even for the most loaded case corresponding to 3 consumers and 30 Interests per second, the mean delay stays well below 50 ms.

Consider now the share of received compute results in time jointly characterizing reliability and full delay in Figure 10 for the same input parameters of the system. As one may observe, in all the considered cases, all the requests are satisfied as the metric of Interest eventually approaches one. The most pronounced effect is caused by increasing the number of consumers from one to three. The rationale is that this increases the overall network load, thus leading to higher latency. Still, even in the most loaded considered case with three consumers and intensities of 30 Interests per second, all the requests are served in just 70 ms.

#### 4.3.2. Mobile Mesh Environment

Having analyzed compute performance in static mesh conditions, we next explore the impact of adding mobility to the nodes on mesh dynamics. Figure 11 illustrates the fraction of satisfied Interests as well as mean full delay for different speeds of nodes, 5×5 mesh, enabled cache, and one and three producers. Observing the reported data, one may conclude that even slight mobility leads to drastic performance degradation in terms of satisfied fraction of Interest. The change between v=0 m/s and v=1 m/s is abrupt, decreasing the amount of satisfied Interest by 90% and 80% for M=3 and M=1, respectively. The major source of performance degradation is related to link breaks between end nodes, leading to the eventual loss of packets. It is important to note that the increase in the number of producers has a positive effect on the system’s performance as more nodes have the requested service available.

Notably, the mean full delay of delivered packets remains almost intact, with mobility coming into play. The rationale is that in mobile meshes, the considered compute service becomes opportunistic, and the Interests are fulfilled only when there is connectivity between nodes at the moment of request. The trade-off between delay and reliability is further illustrated in Figure 12. Please note that the considered characteristics improve in denser mesh conditions as the probability of the node being disconnected from the network decreases. However, these gains are further reduced by the increased interference level. It is important to note that the caching capability inherent in NDN technology does not significantly enhance compute service performance in dynamic meshes, although it is likely to be influenced by cache parameters such as size and freshness.

#### 4.3.3. Effects of Traffic Type and Caching

One of the key benefits of NDN systems is the implementation of caching at intermediate nodes. We now examine how caching and different types of traffic impact compute service performance in a static mesh environment. To this effect, Figure 13 illustrates the share of received compute results in time for 5×5 mesh configuration, with cache enabled (*C*) and disabled (*N*) as a function of the number of consumers and a single producer. Here, we consider two traffic types, CBR and Zipf. It is important to remember that CBR uniformly selects the compute service from all available services, whereas Zipf traffic reflects content popularity, arranging request probabilities according to service popularity as modeled by the Zipf distribution [51].

From the data presented in Figure 13, one may confirm that for both considered traffic types, the share of the received compute results reaches 100% irrespective of whether a cache is enabled or not. Analyzing the presented data further, one may conclude that the effect of cache is non-uniform across other service parameters. In particular, for just a single consumer, the effect of caching is insignificant. However, as the number of consumers increases (and thus the intensity of Interests), caching plays a noticeable role. Notably, the gain between K=3 and K=1 approaches 0.3 for 50 ms, i.e., 0.6 vs. 0.9 share of the satisfied Interests.

A cross-comparison between Figure 13a and Figure 13b reveals that the caching effect is significantly more evident for the Zipf traffic. The rationale is that this traffic type results in a much wider spread of popular content among the mobile nodes. Remarkably, the difference between enabled and disabled cache is evident already for K=1 for the Zipf traffic model. Similarly, it is easy to observe the reliability and delay gains of Zipf traffic over the CBR traffic.

#### 4.3.4. Effects of Network Size and Nodes Density

We finalize the exposure of our mechanisms by comparing the performance of the considered dynamic compute service in two extreme cases: large mobile mesh featuring 49 nodes and small static mesh consisting of just 25 (5×5) and 9 (3×3) nodes in Figure 14 for a range of input system parameters. Please note that in both cases, the node’s density is kept constant. As one may observe by analyzing the presented data, the amount of satisfied Interests barely reaches 0.3 for a large mobile mesh within 200 ms. This behavior is regulated by two factors: (i) high interference conditions and (ii) events of nodes’ disconnections from the network. Although there are no measures to combat the latter effect, one may utilize spatial frequency reuse in the underlying wireless technology to improve the performance of the system. In the same interval, is 5×5 and 3×3 meshes, all the Interests are successfully served.

As one may observe, the performance of the dynamic compute service heavily depends on the network size. To investigate this effect, deeply revealing and isolating the sources of losses, we complement the previous results by assessing the impact of network density and frequency of requests. To this end, Figure 15 presents satisfied Interest fraction and mean delay for various node velocities in mobile mesh for different initial spacing between nodes specifying the compartment size and thus directly affecting node density. Analyzing the presented results, one may observe that the interplay between node density and velocity is characterized by a complex behavior. In particular, higher velocity leads to worse performance for the lowest considered density. The rationale is that for this density, higher velocity frequently leads to situations where nodes lose entirely network connectivity. Thus, the increase in the network density leads to better performance for higher velocities and worse performance for lower speeds, i.e., for initial internode spacing of 40 m, the satisfied Interest fractions for different velocities even out. This is explained by the second phenomenon affecting system performance—interference. Indeed, by increasing the network density, the amount of interference in the system increases, reducing the satisfied Interest fraction.

Figure 16 illustrates the satisfied Interest fraction and mean delay for 10 Interest/s frequency, different initial internode spacing, and different nodes’ velocities. By directly comparing the results of Figure 16 and those from Figure 15 illustrating the same metrics for 1 Interest/s, one may observe that the increase in the traffic load logically worsens the system performance for all considered values of nodes velocities and density. The effect of the increased interference impacts the considered deployments differently. In particular, it affects denser deployments with high node velocities, which are much heavier compared to sparser deployments with lower node velocities.

#### 4.3.5. Advanced Networking Mechanisms

Please note that the performance of the proposed dynamic compute service may also heavily depend on underlying technology parameters. To this aim, Figure 17 quantifies the satisfied Interest fraction for different WiFi parameters, including disabled and enabled RTS/CTS (denoted by “RTS” and “No RTS”) functionalities, single and multi-channel operational modes (denoted by “S” and “M”) for mobile and static meshes (denoted by “Mob.” and “Stat.”). Recall that previously, we assumed that RTS/CTS functionality was disabled while all nodes operated using the same channel.

Analyzing the results presented in Figure 17, one may observe that the use of the RTS/CTS scheme does not drastically affect system performance. The only case where it provides a noticeable effect is the case of static environment with single channel operation. The reason for this is that in these conditions, the major source of incorrect packet reception is interference. Enabling RTS/CTS functionality allows improving performance. When nodes are permitted to utilize multiple channels via adaptive channel selection functionality, the interference is negligible, and thus no gains are evident. Addressing the effect of adaptive channel selection, we would like to note that it produces the most effect in static conditions when RTS/CTS is disabled. In mobile conditions, most of the losses are mainly caused by node mobility and, thus, not affected by this functionality.

## 5. Conclusions

Aiming for the dynamic computing service, in this paper, we extended NDN functionality to the wireless meshes. In particular, we proposed the dynamic face management system enabling unicast transmissions over an inherently broadcast medium and a novel learning-based Self-Learning Strategy that allows a decrease in the number of signaling packets distributed in the network to find paths to the sources. The former allows for unicast transmission, thus reaching the full potential of wireless technologies, e.g., WiFi, while the latter ensures that the amount of overhead for content search is minimized. Both features are specifically useful in dynamic wireless environments such as mesh systems, where the content location and as well as nodes’ connectivity may change as a result of topology changes. Furthermore, to facilitate dynamic compute service, we utilized multiple packet response functionality over NDN from [23]. Our modifications are inherently compatible with the NDN concept and do not alter its core functionality.

We then proceeded to evaluate the effects of dynamic compute service over dynamic meshes. Our findings reveal that for static meshes, our proposal allows us to fully benefit from the computing resources available sporadically around the consumer. However, similar to IP-based networks, mobility is still the most dominating source of performance degradation. In a mobile mesh environment, the performance is determined by the interplay between node velocity, deployment density, and traffic intensity, i.e., higher velocity improves performance in denser scenarios and deteriorates in sparser deployments. The inherent caching capability of NDN technology does not help to improve the performance of mobile systems. Denser conditions do not allow for improving the system’s performance as the level of interference increases. Finally, we characterized the effects of different types of traffic and NDN caching capabilities by showing that the latter results in much better system performance while the popularity of the compute service adds additional performance gains. The underlying wireless technology functionality, such as RTS/CTS, may also affect service performance in specific conditions.

In addition, a particularly promising direction for expanding our work is the incorporation of smart name aggregation into more generic prefixes over time. This strategy can effectively limit the required memory for storing local routing information. By adopting such an approach, our system could achieve greater efficiency and scalability, especially in scenarios where resources are constrained. This enhancement not only optimizes our current framework but also broadens the applicability of our system in various wireless mesh network environments [52]. Finally, we would like to mention that the proposed and the current Face implementations introduce different types of overheads in dynamic network conditions. The comparison of these overheads, as well as full performance evaluation and comparison between current and suggested implementations, are a part of our future work.

## Figures and Tables

**Figure 1 sensors-24-01120-f001:**
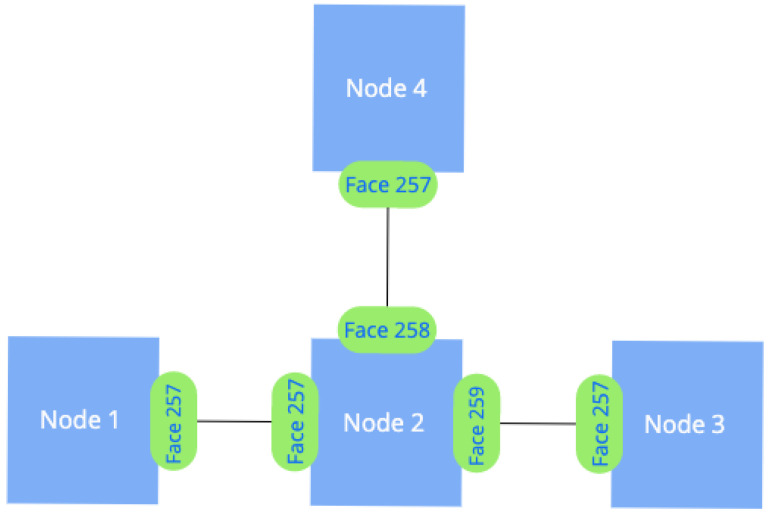
Utilization of Faces in wired scenarios.

**Figure 2 sensors-24-01120-f002:**
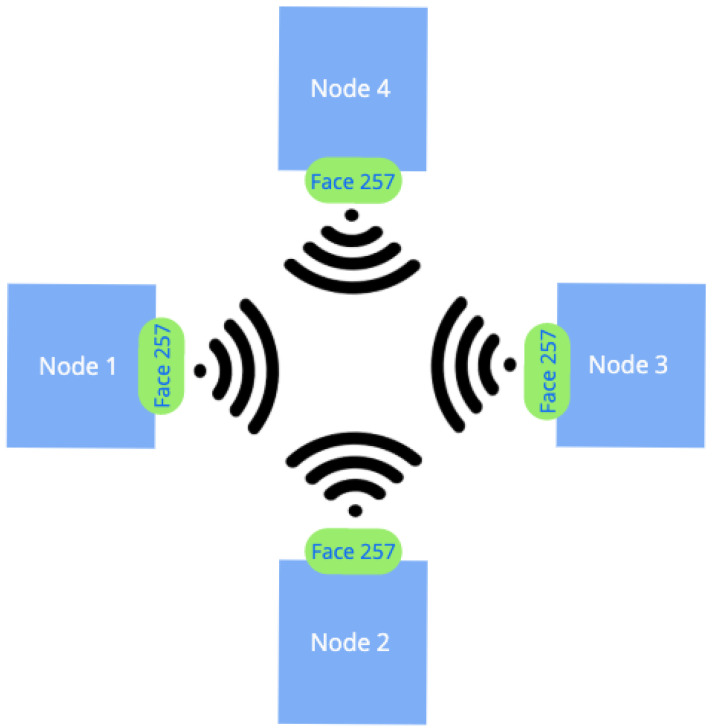
Utilization of Faces in wireless scenarios.

**Figure 3 sensors-24-01120-f003:**
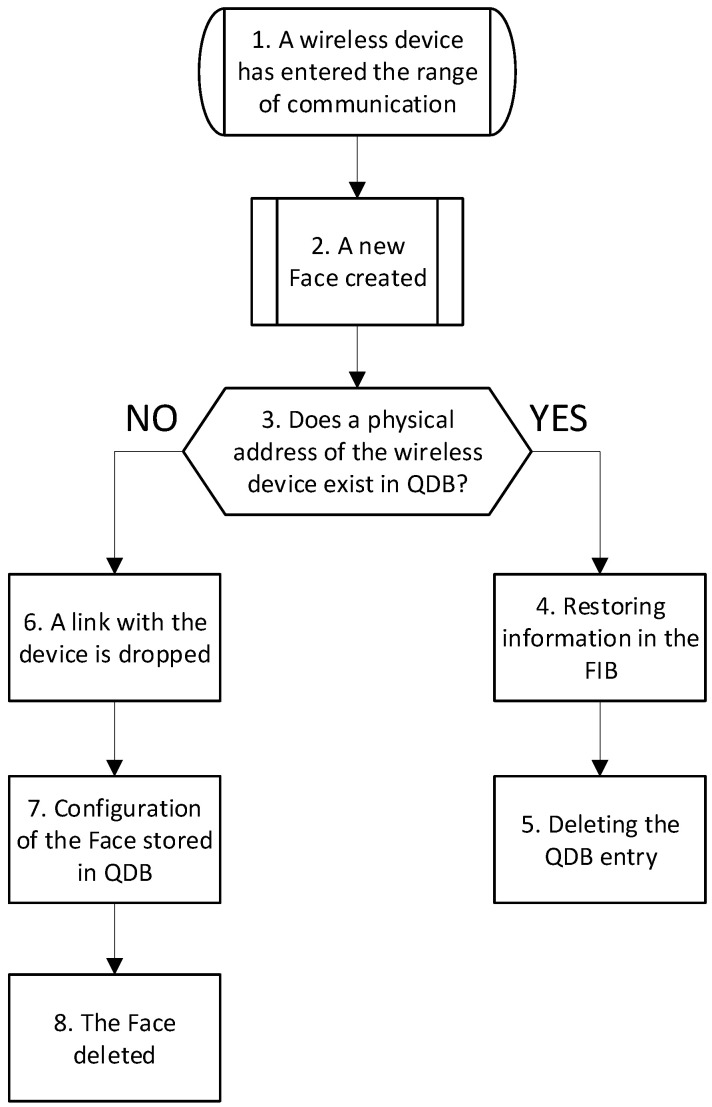
Workflow of the proposed solution.

**Figure 4 sensors-24-01120-f004:**
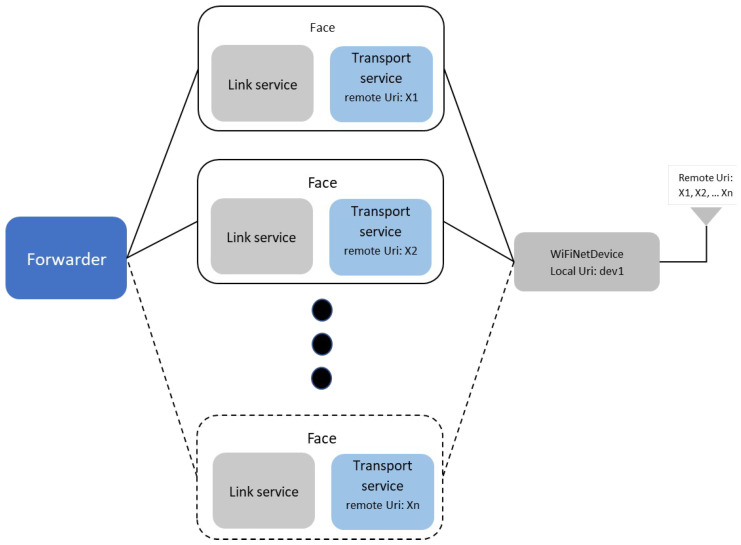
Illustration of the proposed modifications.

**Figure 5 sensors-24-01120-f005:**
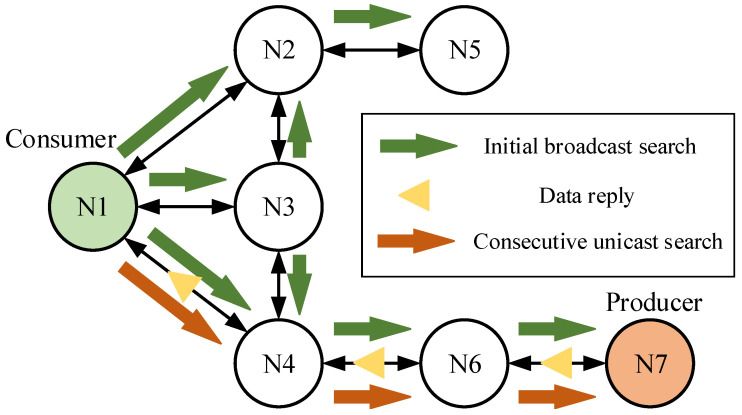
Illustration of the proposed window forwarding strategy.

**Figure 6 sensors-24-01120-f006:**
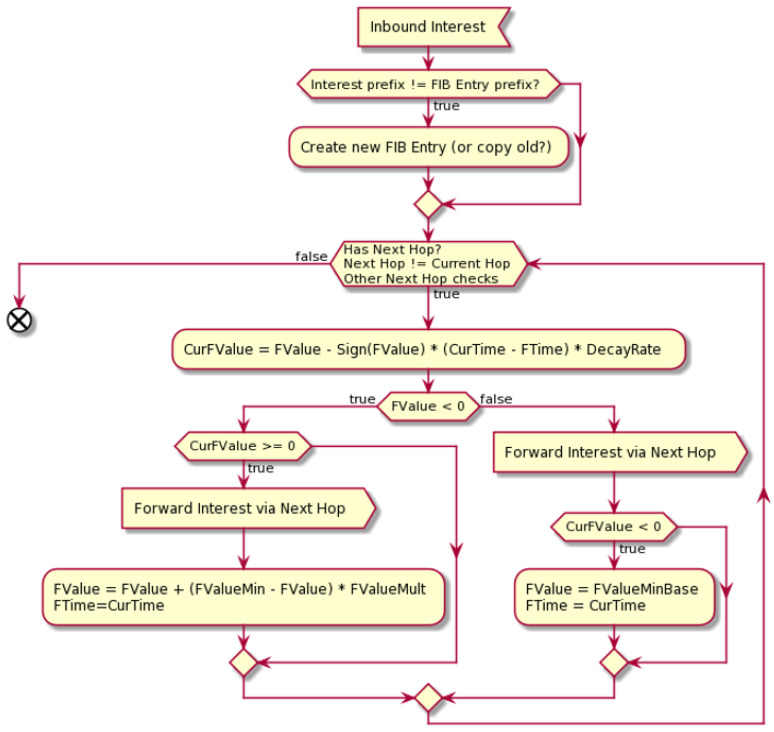
Inbound Interest handling.

**Figure 7 sensors-24-01120-f007:**
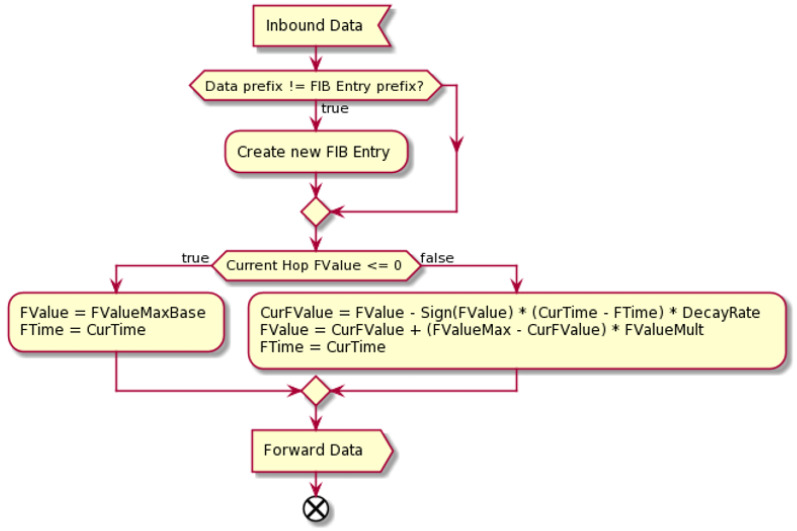
Inbound Data handling.

**Figure 8 sensors-24-01120-f008:**
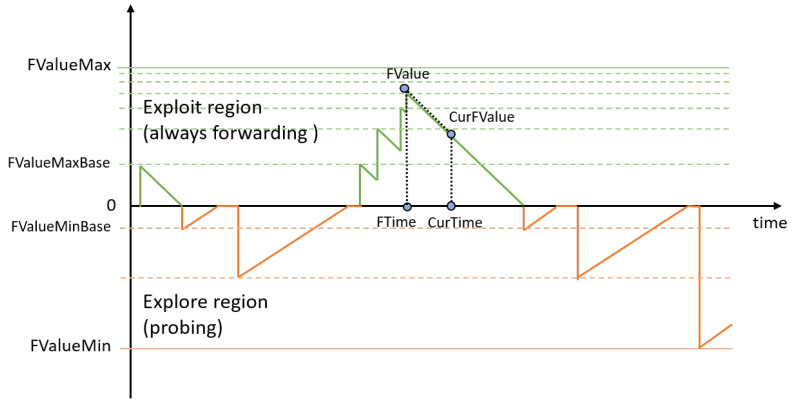
Example of the forwarding time window dynamics.

**Figure 9 sensors-24-01120-f009:**
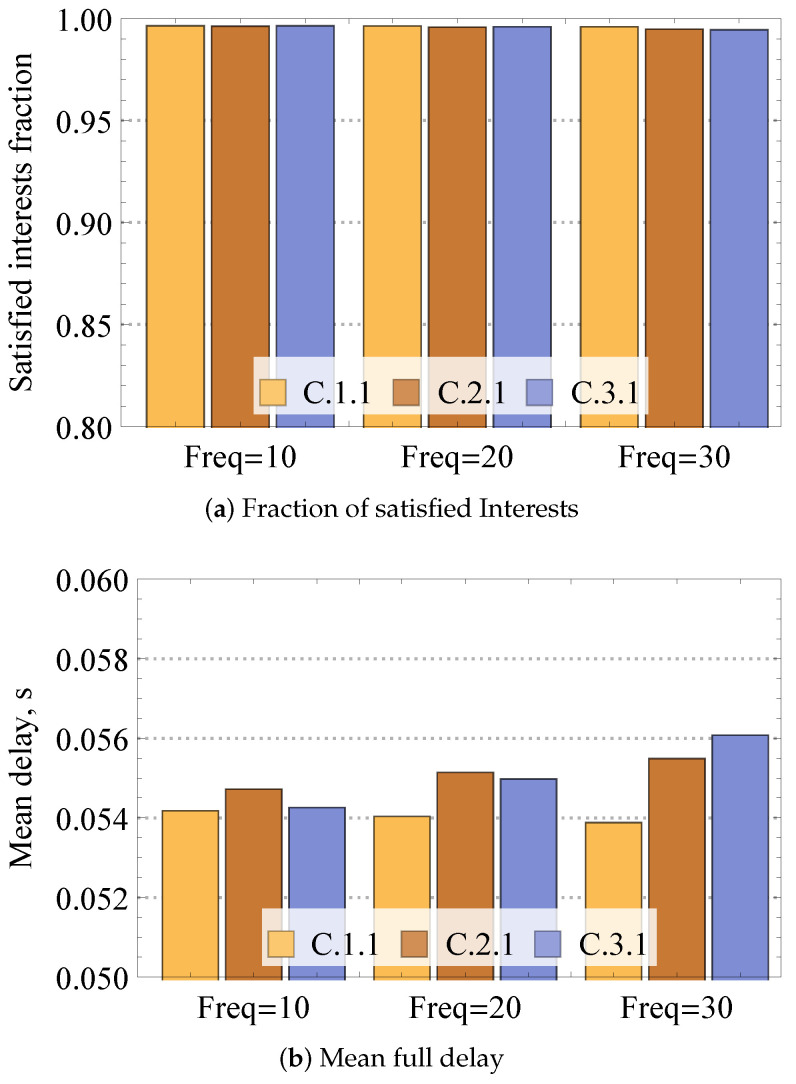
Loss and delay performance measures in static mesh.

**Figure 10 sensors-24-01120-f010:**
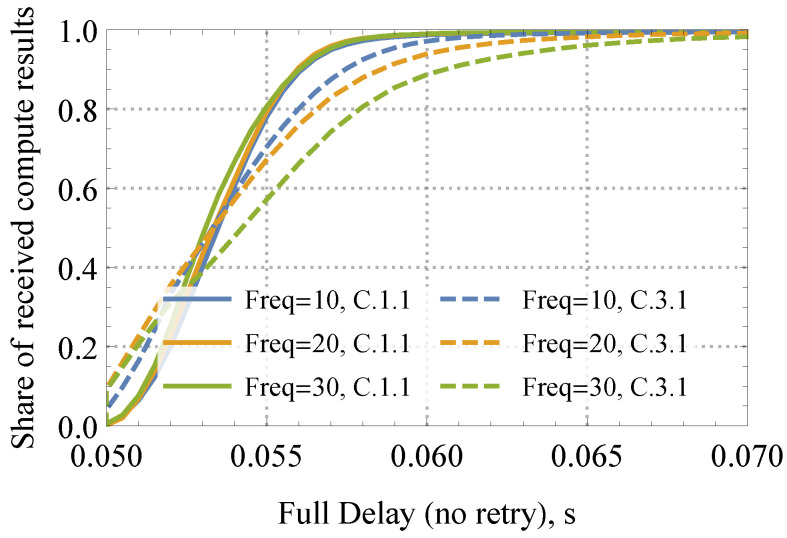
Share of received compute results in static mesh.

**Figure 11 sensors-24-01120-f011:**
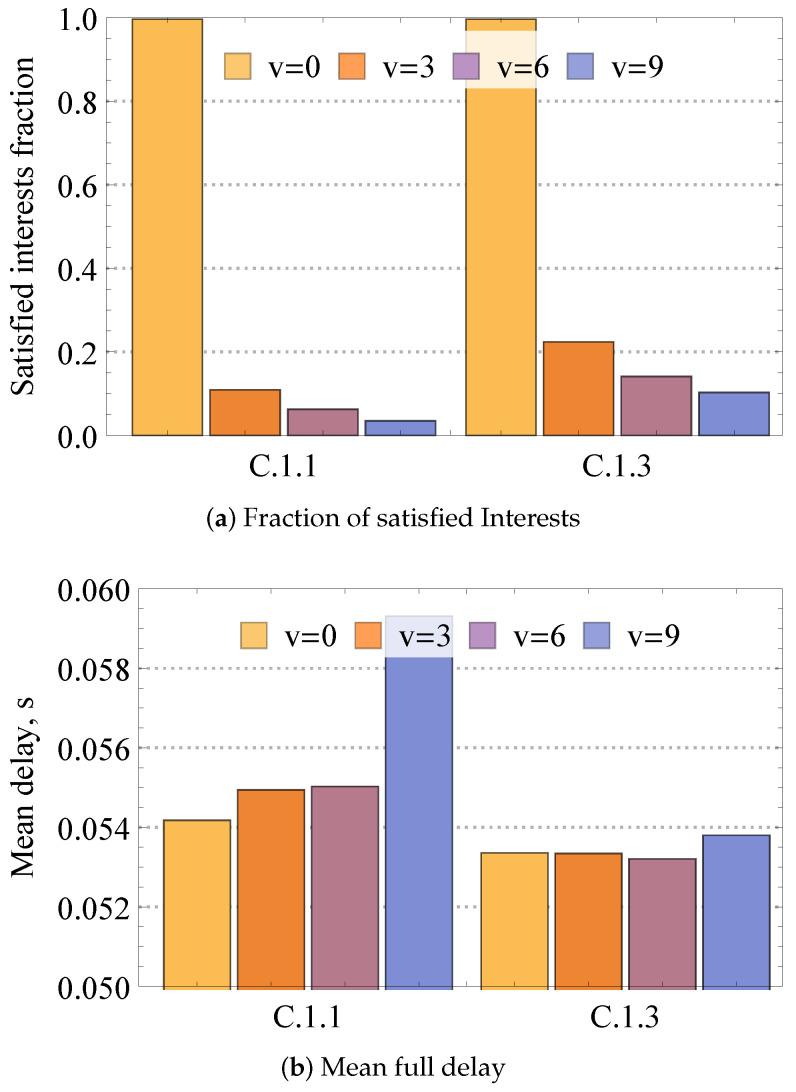
Loss and delay performance measures in mobile mesh.

**Figure 12 sensors-24-01120-f012:**
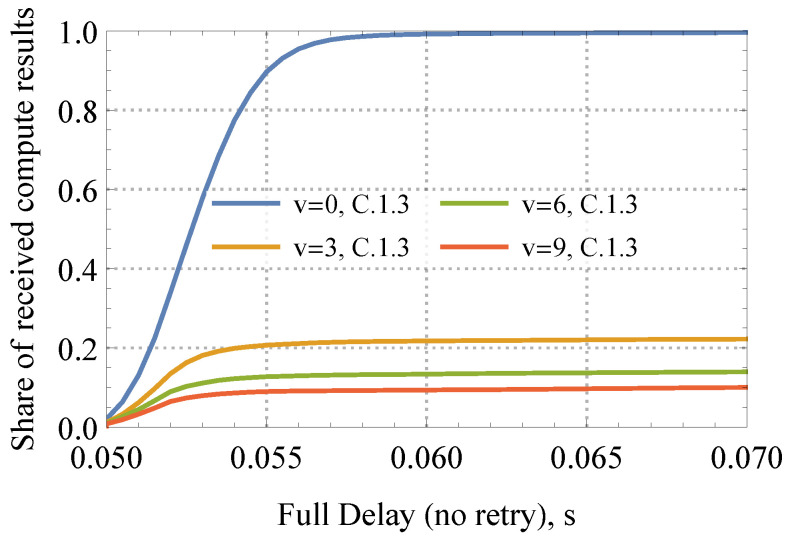
Share of received compute results in a mobile mesh.

**Figure 13 sensors-24-01120-f013:**
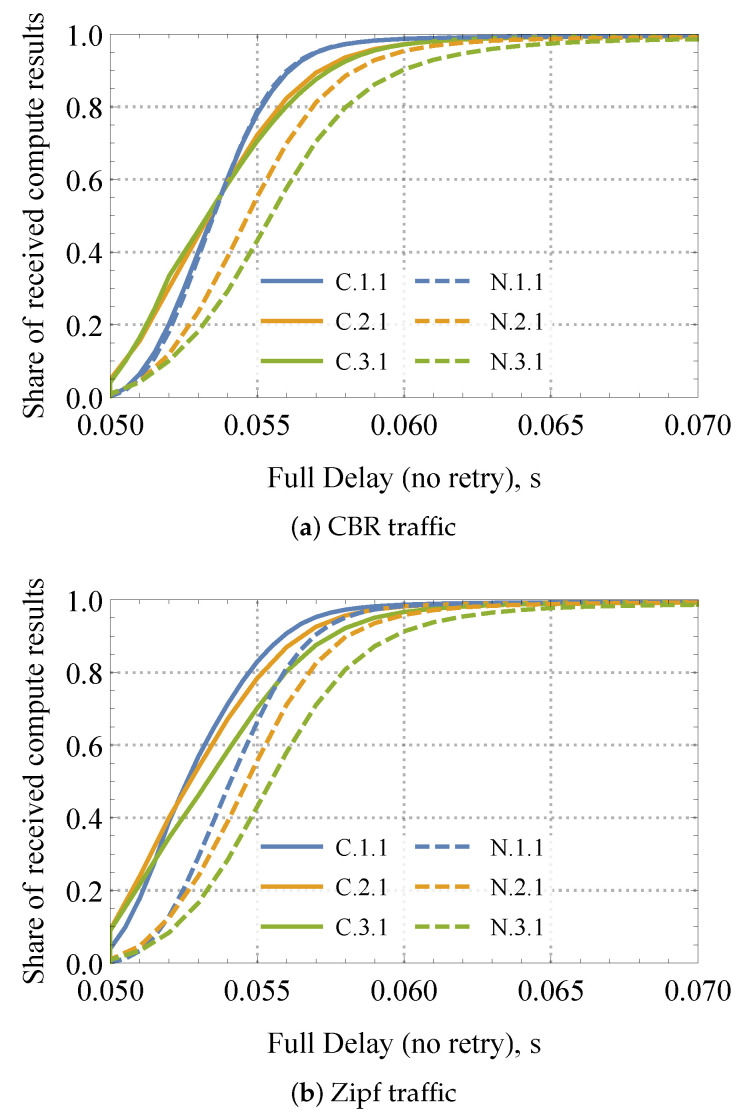
Loss and delay performance measures in mobile mesh.

**Figure 14 sensors-24-01120-f014:**
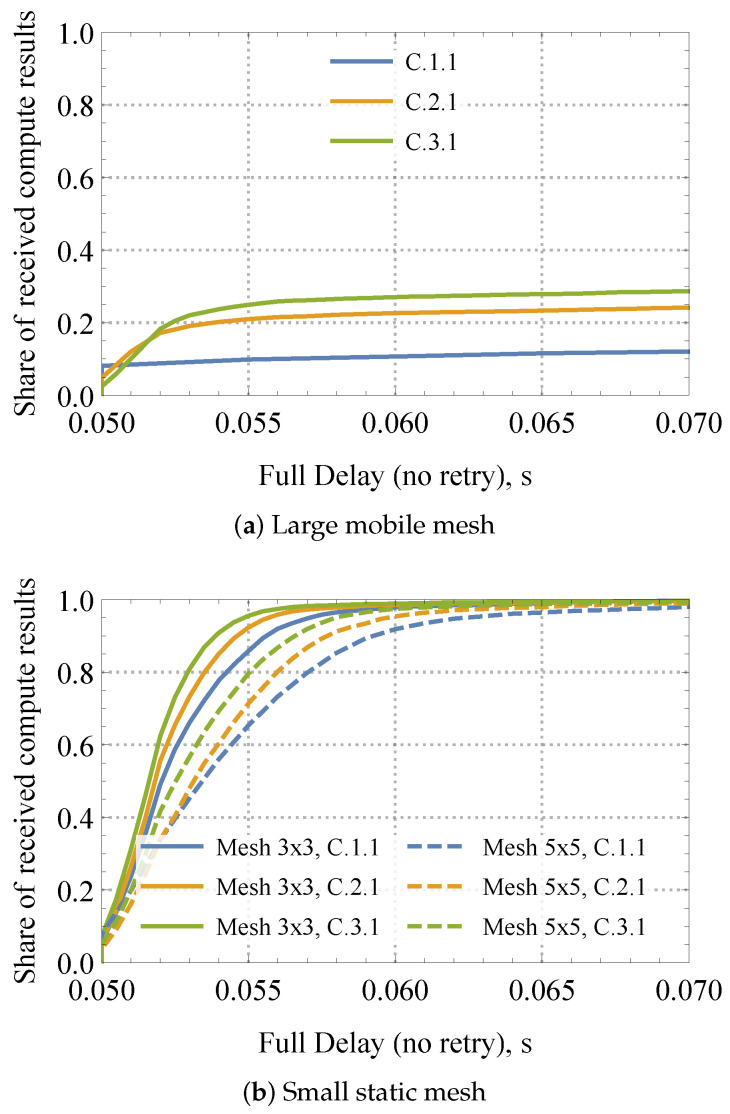
Loss and delay performance measures in mobile mesh.

**Figure 15 sensors-24-01120-f015:**
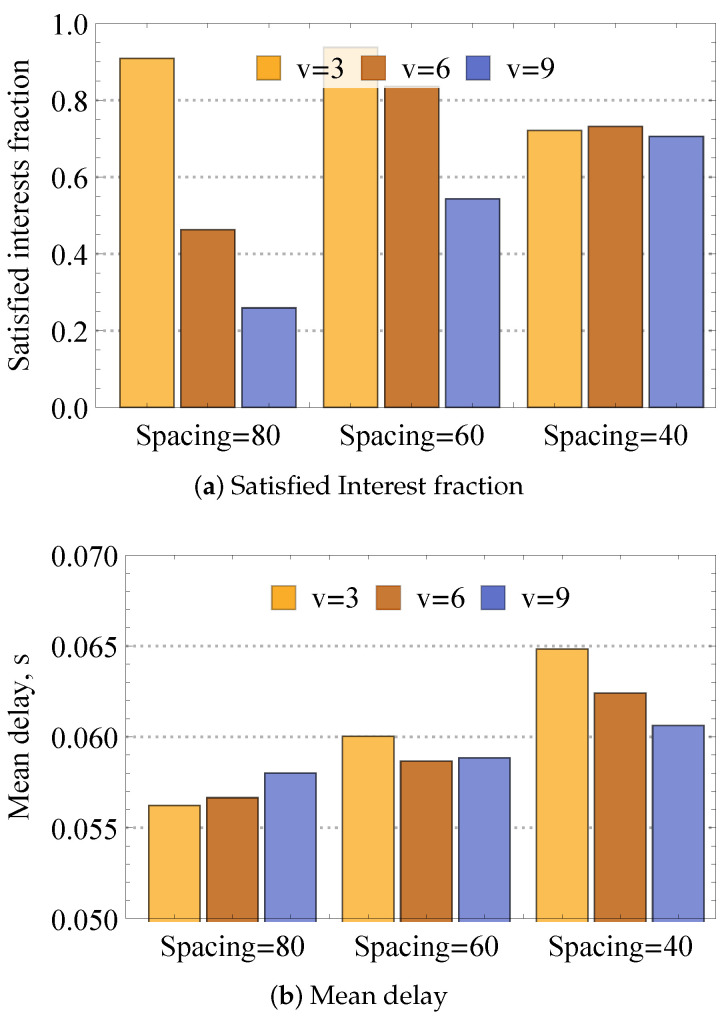
Loss and delay metrics for different network densities.

**Figure 16 sensors-24-01120-f016:**
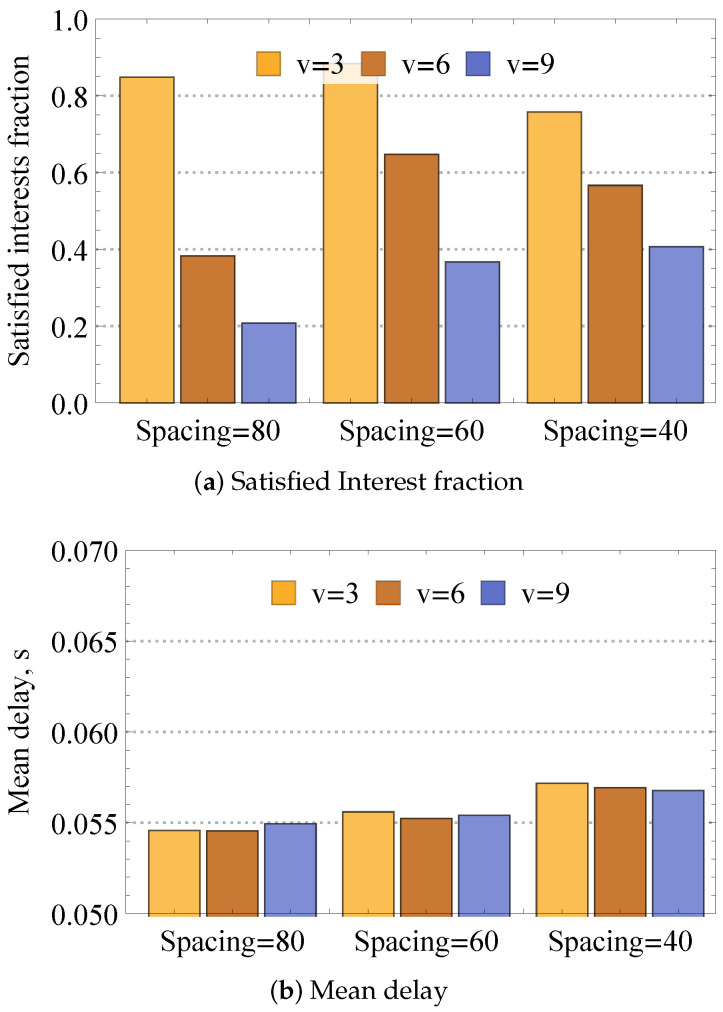
Loss and delay metrics for 10 Interest/s frequency.

**Figure 17 sensors-24-01120-f017:**
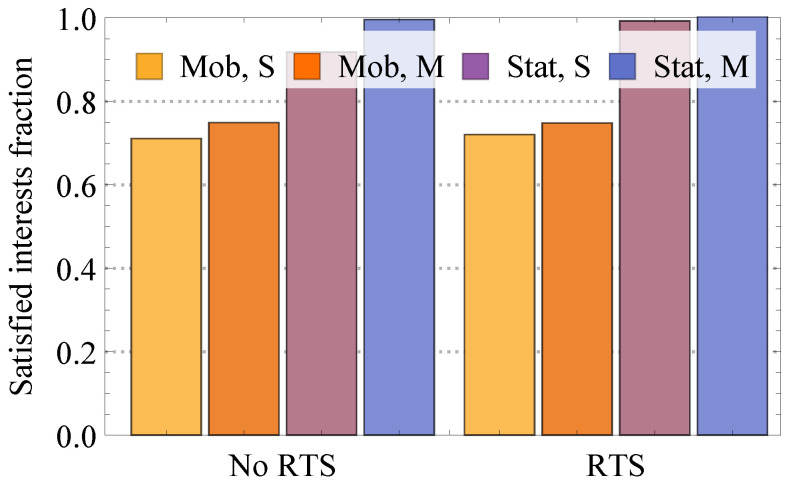
Satisfied Interest fraction.

**Table 1 sensors-24-01120-t001:** Abbreviations used in the considered self-learning strategy algorithm.

Label	Description
FValue	Forwarding coefficient associated with every available next hop
CurFValue	Value of forwarding coefficient at the current moment of time
CurTime	Current moment of time
FTime	A moment of time when a last Interest with a certain prefix was forwarded via the Face
DecayRate	Coefficient regulating change in FValue in time
FValueMin	Minimal possible value of FValue
FValueMinBase	Starting negative value assigned to FValue when associated Face has entered the explore region
FValueMax	Maximal possible value of FValue
FValueMaxBase	Starting positive value assigned to FValue when associated Face has entered the exploit region
FValueMult	Coefficient regulating change in FValue with every forwarding attempt

**Table 2 sensors-24-01120-t002:** Default parameters for simulation campaign.

Parameter	Value
Runs for each experiment	100
Number of wireless nodes, *N*	$3^{2}$, $5^{2}$, $7^{2}$ grids
Mobility model	Random direction model [48]
Wireless technology	IEEE 802.11n
Initial internode distance, *L*	95 m
Frequency band	5 GHz ISM band
Propagation model	FSPL
Coverage range	100 m
RTS/CTS operation	Disabled
Automatic frequency selection	Disabled
Velocity of wireless nodes, *V* m/s	0, 3, 6, 9
NDN application type	CBR, Zipf
Cache type	LRU
Cache size, compute results	20 units
Interest frequency	1 Interest/s
Results freshness	1000 ms
Compute time	1 ms
Server selection time	6 ms
Number of consumers	{1, 3} nodes
Number of producers	{1, 3} nodes

## Data Availability

Data are contained within the article.

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
