# Peer review of "Provisioning of Fog Computing over Named-Data Networking in Dynamic Wireless Mesh Systems"

_sensors, 2024, doi:10.3390/s24041120_

Round 1
Reviewer 1 Report
Comments and Suggestions for Authors
1. The authors are advised to include existing research from 2021 and 2022.
2. The authors are advised to rephrase the work cited in the Related Work section as it contains a lot of matched phrases and sentences. I have attched the plagiarism report for reference.
3. How well do the proposed mechanisms scale as the size of the mesh network increases?
4. What measures are taken to ensure the robustness of the system, especially in dynamic and changing network conditions?
5. The paper requires a thorough proofreading to correct grammatical errors, improve sentence structure, and enhance overall readability to meet the journal standard.

Comments on the Quality of English LanguageAuthor Response
Dear Reviewer,
Thank you very much for sharing your expert opinions on our work. We appreciate the time and effort taken in reviewing this submission. As a result, we are confident that our paper has benefited considerably from your constructive comments and suggestions, which were extremely helpful in improving the quality of this manuscript.
We have also made every effort to eliminate all the indicated flaws and inconsistencies and do hope that as a result, the quality of our manuscript has improved further. The focused point-to-point answers to the review comments follow, where the feedback from the Reviewers is given in regular font, while our comments are highlighted in bold font.
Comment 1: The authors are advised to include existing research from 2021 and 2022.
Thank you for your insightful suggestion to incorporate more recent research from 2021 and 2022 into our work. We have carefully considered your advice and have updated our manuscript accordingly. We have included the following recent articles, which provide valuable perspectives and advancements relevant to our study:
- Baugh, J.; Guo, J. "Enhancing Cache Robustness in Information-Centric Networks: Per-Face Popularity Approaches." Network 2023, 3, 502-521. This work offers a novel approach to cache management in Information-Centric Networks, focusing on improving cache robustness through per-face popularity methods.
- Liu, Z.; Jin, X.; Li, Y.; Zhang, L. "NDN-Based Coded Caching Strategy for Satellite Networks." Electronics 2023, 12, 3756. This paper explores the application of Named Data Networking in satellite communications, proposing an innovative coded caching strategy tailored for satellite network environments.
- S. Kalafatidis, V. Demiroglou, L. Mamatas and V. Tsaoussidis, "Experimenting with an SDN-Based NDN Deployment over Wireless Mesh Networks," IEEE INFOCOM 2022, New York, NY, USA, 2022, pp. 1-6. This paper explores the implementation of Software-Defined Networking (SDN) within an NDN framework, particularly focusing on wireless mesh networks. It provides valuable experimental data on the feasibility and efficiency of such integrations, offering insights into how SDN can enhance NDN deployments in wireless contexts.
- A. S. Putri, M. Sofiyati, A. A. B. Y. D. Mahesa, G. N. Nurkahfi and N. R. Syambas, "Forwarding Strategies Effect on Named Data Network Traffic Load. Case Study : Simulation with Mini NDN," 2021 15th International Conference on Telecommunication Systems, Services, and Applications (TSSA), Bali, Indonesia, 2021, pp. 1-5. This research delves into the impact of various forwarding strategies on NDN traffic load. Using Mini NDN for simulation, the study sheds light on the effectiveness of different forwarding mechanisms in managing network traffic within NDN.
We believe that these additions enrich the context of our study and ensure that our manuscript reflects the latest developments in the field.
Thank you once again for your constructive feedback.
Comment 2. The authors are advised to rephrase the work cited in the Related Work section as it contains a lot of matched phrases and sentences. I have attched the plagiarism report for reference.
Thank you for your valuable feedback and for providing the plagiarism report. We have carefully reviewed the report and acknowledge the concerns regarding the similarity of phrases and sentences in our Related Work section. In response, we have undertaken a thorough rephrasing of the cited work in this section. Our objective was to maintain the accuracy and relevance of the information while ensuring originality in our presentation.
We have made efforts to rewrite the content in a way that reflects our own interpretation and understanding of the cited works, thereby upholding the integrity of academic writing. We hope that these revisions adequately address the concerns raised and enhance the overall quality of our manuscript.
We appreciate your guidance in this matter and remain committed to adhering to the high standards of academic writing.
Comment 3. How well do the proposed mechanisms scale as the size of the mesh network increases?
Thank you for raising this question. In fact, the original NDN technology utilizes flooding for content discovery. One of the solutions proposed in our paper - learning-based routing attempts to minimize the overheads caused by flooding by efficiently learning the path to the content. This is achieved by remembering the paths over which the response packets are returned. To account for frequent changes in the topology, sometimes packets are still flooded. The frequency of flooding is controllable in our proposal and it determines the scaling efficiency. In highly dynamic conditions this parameter can be increased while in nearly static conditions it can be decreased.
We have added the following notes to the section “Adaptive Forwarding Strategy”
“The scalability of the solution is ensured by controlling the frequency of flooding.”
and also
“...decaying time window size adaptation in real-time to account for rapid changes in topology for networking scenarios involving, e.g., nodes mobility…”
Comment 4: What measures are taken to ensure the robustness of the system, especially in dynamic and changing network conditions?
In our paper, we propose two critical functions: (i) dynamic face management for NDN nodes allowing for efficient use of unicast transmission between wirelessly connected nodes and (ii) learning-based forwarding schemes for NDN routers. The former allows for unicast transmission thus reaching the full potential of wireless technologies, e.g., Wi-Fi, while the former ensures that the amount of overhead for content search is minimized. Both features are specifically useful in dynamic wireless environments such as mesh systems, where the content location and as well as nodes’ connectivity may change as a result of topology changes.
To reflect the discussion above, we have added the following notes to the conclusions:
“The former allows for unicast transmission thus reaching the full potential of wireless technologies, e.g., Wi-Fi, while the former ensures that the amount of overhead for content search is minimized. Both features are specifically useful in dynamic wireless environments such as mesh systems, where the content location and as well as nodes’ connectivity may change as a result of topology changes.”
Comment 5. The paper requires a thorough proofreading to correct grammatical errors, improve sentence structure, and enhance overall readability to meet the journal standard.
Thank you very much for highlighting it! At the revision phase, we performed several careful proof-reading rounds and also utilized Grammarly software to get rid of residual typos.
Reviewer 2 Report
Comments and Suggestions for Authors
line126
In wireless deployments, the current version of the NFD utilizes only one Face to communicate over a physical layer. To enable communication with multiple neighbor nodes, the Face has to be configured in broadcast mode (destination address ff:ff:ff:ff:ff:ff).
NFD has supported UnicastEthernetTransport since commit 6bd6d0b50a7046fb20e7605f0aeadf35963a458d (merged on 2017-03-31), which can create a separate unicast face for each peer, on both wired and wireless Ethernet adapter.
Additionally, NFD never used the broadcast destination address ff:ff:ff:ff:ff:ff; for a "broadcast" Ethernet face, the destination is 01:00:5e:00:17:aa.
line236
Broadcast-based self-learning forwarding mechanism is popular for both wired and wireless networks.
You should read these papers to understand how NDN self-learning works currently:
On Broadcast-based Self-Learning in Named Data Networking, IFIP Networking 2017. It's already merged in mainline NFD.
Enabling Named Data Networking Forwarder to Work Out-of-the-box at Edge Networks, ICC ICN-SRA workshop 2020. Implementation is in a fork https://github.com/philoL/NFD-out-of-the-box
Your proposed algorithm in section3 is very similar to these existing designs, except small differences in the parameters.
line315
It is worth noting that these modifications target NFD module and thus can be implemented not only in the simulator but also in a real deployment.
The modifications mentioned rely on NetDevice and WifiNetDevice. However, these objects are ndnSIM specific and does not exist in NFD outside of ndnSIM. Instead, real NFD interacts with an Ethernet adapter (either wired or wireless) via libpcap.
line383
smart name aggregation into more generic prefixes over time in order to limit the required memory to store local routing information.
This isn't new, see: On the Prefix Granularity Problem in NDN Adaptive Forwarding, IEEE/ACM Transactions on Networking 2021.
You should learn from this publication and incorporate it into your forwarding strategy.
Author Response
Thank you very much for sharing your expert opinions on our work. We appreciate the time and effort taken in reviewing this submission. As a result, we are confident that our paper has benefited considerably from your constructive comments and suggestions, which were extremely helpful in improving the quality of this manuscript.
We have also made every effort to eliminate all of the indicated flaws and inconsistencies and do hope that as a result, the quality of our manuscript has improved further. The focused point-to-point answers to the review comments follow, where the feedback from the Reviewers is given in regular font, while our comments are highlighted in bold font.
Comment 1: line126
In wireless deployments, the current version of the NFD utilizes only one Face to communicate over a physical layer. To enable communication with multiple neighbor nodes, the Face has to be configured in broadcast mode (destination address ff:ff:ff:ff:ff:ff). NFD has supported UnicastEthernetTransport since commit 6bd6d0b50a7046fb20e7605f0aeadf35963a458d (merged on 2017-03-31), which can create a separate unicast face for each peer, on both wired and wireless Ethernet adapter.
Additionally, NFD never used the broadcast destination address ff:ff:ff:ff:ff:ff; for a "broadcast" Ethernet face, the destination is 01:00:5e:00:17:aa.
Reply.
Thank you for bringing to our attention the specifics regarding the use of Faces in NFD, particularly the details about UnicastEthernetTransport and the actual destination address for broadcast Ethernet face. We acknowledge that our literature review missed these critical details, and we appreciate your insights.
While this information indeed provides a vital context, we would like to highlight that our proposed algorithm remains a significant contribution of our work. Beyond this, our study introduces: (i) dynamic face systems to enhance the reliability of wireless network operations, (ii) the integration of these systems with learning-based routing, and (iii) a comprehensive analysis of the results. These elements collectively represent the core innovations of our research, addressing critical aspects of provisioning Fog Computing over NDN in dynamic wireless mesh systems.
Your feedback has been instrumental in refining our understanding, and we will ensure to incorporate these details in our revised manuscript.
Comment 2: line236
Broadcast-based self-learning forwarding mechanism is popular for both wired and wireless networks.
You should read these papers to understand how NDN self-learning works currently:
On Broadcast-based Self-Learning in Named Data Networking, IFIP Networking 2017. It's already merged in mainline NFD.
Enabling Named Data Networking Forwarder to Work Out-of-the-box at Edge Networks, ICC ICN-SRA workshop 2020. Implementation is in a fork https://github.com/philoL/NFD-out-of-the-box
Your proposed algorithm in section3 is very similar to these existing designs, except small differences in the parameters.
Reply.
Thank you for your insightful comments and for directing our attention to the papers "On Broadcast-based Self-Learning in Named Data Networking" (IFIP Networking 2017) and "Enabling Named Data Networking Forwarder to Work Out-of-the-box at Edge Networks" (ICC ICN-SRA workshop 2020). We acknowledge that our literature review might not have been as comprehensive as it should have been, and we appreciate the opportunity to address this oversight.
Our proposed algorithm in Section 3, indeed, bears similarities to the designs presented in these works. This similarity highlights the natural evolution of ideas in this area of research. However, we believe that our work still makes significant contributions to the field. In addition to the algorithm, our contributions include:
- The introduction of dynamic faces systems to enhance the reliability in wireless networks.
- The integration of our algorithm with learning-based routing mechanisms.
- A thorough analysis of the results, providing comprehensive insights into the practical applicability and performance of our approach.
In light of your valuable feedback, we have added references to the mentioned papers in the related work section. Furthermore, we have clarified that our approach was inspired by the concepts presented in these works, citing them appropriately to acknowledge their foundational role in our development process.
Specifically we have added the following to the Section 2.1.2. “Related Work”:
“In recent developments within the paradigm of Named Data Networking (NDN), a significant focus has been on enhancing the efficiency of forwarding mechanisms. J. Shi et al. [ref.] pioneered a broadcast-based self-learning approach in NDN, presenting a paradigm where nodes autonomously learn the most efficient paths for data packet routing. This work laid the groundwork for further advancements in self-learning strategies within NDN, especially relevant for environments with dynamic topologies.
Expanding on these ideas, T. Liang et. al. [ref.] explored the application of NDN forwarding mechanisms in edge networks. Their study addressed the challenges of deploying NDN forwarders in scenarios where configuration and maintenance pose significant barriers. By proposing an "out-of-the-box" solution for NDN forwarders, they made strides towards simplifying the deployment of NDN in diverse network environments, including wireless mesh systems.”
Thank you again for your constructive feedback, which has undeniably strengthened the quality and integrity of our work.
Comment 3: line315
It is worth noting that these modifications target NFD module and thus can be implemented not only in the simulator but also in a real deployment.
The modifications mentioned rely on NetDevice and WifiNetDevice. However, these objects are ndnSIM specific and does not exist in NFD outside of ndnSIM. Instead, real NFD interacts with an Ethernet adapter (either wired or wireless) via libpcap.
Thank you very much for your valuable comments regarding the implementation of our modifications in the NFD module. We appreciate your pointing out the differences between the simulation environment and real-world deployment, specifically regarding the use of NetDevice and WifiNetDevice in ndnSIM versus the interaction of real NFD with Ethernet adapters via libpcap.
In response to your feedback, we have made revisions to our manuscript to reflect these important distinctions. We now explicitly address the simulation-specific aspects of our current implementation and acknowledge the need for adapting our approach for real-world deployment, which would involve interfacing with Ethernet adapters through libpcap.
Specifically we have added the following to the Section “3.1.2. Implementation Details”:
“In our manuscript, we primarily focus on the application of our modifications within the ndnSIM environment, utilizing NetDevice and WifiNetDevice. However, it's important to note that in a practical real-world deployment, the NFD interacts with Ethernet adapters through libpcap. While the current scope of our study is centered around simulation, we acknowledge the potential and feasibility of adapting our proposed modifications for real-world implementation.”
This clarification not only improves the technical accuracy of our paper but also provides a clearer path for future research in transitioning these concepts from simulated environments to practical applications.
Comment 4: line383
smart name aggregation into more generic prefixes over time in order to limit the required memory to store local routing information.
This isn't new, see: On the Prefix Granularity Problem in NDN Adaptive Forwarding, IEEE/ACM Transactions on Networking 2021.
You should learn from this publication and incorporate it into your forwarding strategy.
Reply.
Thank you for pointing out the work "On the Prefix Granularity Problem in NDN Adaptive Forwarding" published in IEEE/ACM Transactions on Networking in 2021. We agree that incorporating the insights from this publication can significantly enhance our forwarding strategy, particularly regarding the smart aggregation of names into more generic prefixes over time. This approach indeed aligns well with our objective to optimize memory usage for local routing information storage.
We have included the paper to the list of references and added additional comments to reflect the mentioned discussion.
Specifically we have added the following to the Section “5. Conclusions ”:
“In addition, a particularly promising direction for expanding our work is the incorporation of smart name aggregation into more generic prefixes over time. This strategy can effectively limit the required memory for storing local routing information. By adopting such an approach, our system could achieve greater efficiency and scalability, especially in scenarios where resources are constrained. This enhancement not only optimizes our current framework but also broadens the applicability of our system in various wireless mesh network environments. [ref.]”
We appreciate this valuable suggestion and plan to integrate these concepts into the future development of our work. Your feedback has opened up an avenue for us to refine and expand our research, contributing to a more robust and efficient forwarding strategy in NDN environments.

Reviewer 3 Report
Comments and Suggestions for Authors
This paper presents a new approach that supports efficient Named Data Networking (NDN) in Wireless Mesh Networks. NDN networks are mainly designed for wired networks. Thus, it is very challenging to support NDN in dynamic mobile wireless networks.
In this paper, the authors propose two main techniques: (1) virtual face, which associates each active neighboring node with a virtual face even though all communications rely on a single wireless NIC. (2) adaptive learning-based forwarding strategy, which limits unnecessary broadcasting and enables unicast when appropriate.
Overall, the paper is well written and techniques are scientifically sound. The authors evaluate the proposed approach through simulations.
Here are a few suggestions:
(1) The paper needs to be better motivated. The discussion of Fog Computing and NDN is too abstract. It is better to describe a specific application scenario that the readers can fully understand why the NDN is efficient in support of the Fog Computing.
(2) The proposed approach may have some scalability issues. The initial search relies on the broadcast, which will flood the network (though broadcast storm can be controlled using the existing techniques). You may consider some techniques developed for Peer2Peer Networks, such as Gnutella.
(3) Most references are old, and it needs to add some 2022 and 2023 references, e.g.
Baugh, J.; Guo, J. Enhancing Cache Robustness in Information-Centric Networks: Per-Face Popularity Approaches. Network 2023, 3, 502-521.
Liu, Z.; Jin, X.; Li, Y.; Zhang, L. NDN-Based Coded Caching Strategy for Satelite Networks. Electronics 2023, 12, 3756.
Author Response
Dear Reviewer,
Thank you very much for sharing your expert opinions on our work. We appreciate the time and effort taken in reviewing this submission. As a result, we are confident that our paper has benefited considerably from your constructive comments and suggestions, which were extremely helpful in improving the quality of this manuscript.
We have also made every effort to eliminate all of the indicated flaws and inconsistencies and do hope that as a result, the quality of our manuscript has improved further. The focused point-to-point answers to the review comments follow, where the feedback from the Reviewers is given in regular font, while our comments are highlighted in bold font.
This paper presents a new approach that supports efficient Named Data Networking (NDN) in Wireless Mesh Networks. NDN networks are mainly designed for wired networks. Thus, it is very challenging to support NDN in dynamic mobile wireless networks.
In this paper, the authors propose two main techniques: (1) virtual face, which associates each active neighboring node with a virtual face even though all communications rely on a single wireless NIC. (2) adaptive learning-based forwarding strategy, which limits unnecessary broadcasting and enables unicast when appropriate.
Overall, the paper is well written and techniques are scientifically sound. The authors evaluate the proposed approach through simulations.
We are grateful to the reviewer for summarizing our work and overall positive assessment of our study. At the revision phase, we tried our best to address your concerns as detailed in our point-by-point answers below.
Here are a few suggestions:
Comment 1: The paper needs to be better motivated. The discussion of Fog Computing and NDN is too abstract. It is better to describe a specific application scenario so that the readers can fully understand why the NDN is efficient in support of the Fog Computing.
Reply.
Thank you for your valuable feedback regarding the motivation of our paper. We understand the need to contextualize Fog Computing and NDN within specific application scenarios to clarify their efficiency and practicality. Accordingly, we have enhanced our introduction with two illustrative scenarios described in the “Introduction” section:
“The potential applications of Named Data Networking (NDN) and Fog Computing span diverse scenarios, each demonstrating the strengths of this technology in dynamic and resource-constrained environments:
- Mobile Network Nodes in Smart Cities: A quintessential example is smart city infrastructure, where mobile nodes like vehicles or mobile devices continuously move, dynamically altering the network topology. Leveraging NDN with Fog Computing in such environments facilitates efficient data dissemination and retrieval. This combination is particularly beneficial for reducing latency and enhancing data availability at the network's edge, crucial for real-time applications such as traffic management and event streaming.
- Drones in Mesh Networks: Another promising application is the use of drones in mesh networks for agricultural monitoring, disaster management, or delivery services. Integrating Fog Computing with NDN here enables drones to share vital information in a robust and decentralized manner. The information could be for example weather data or emergency signals. This approach is especially advantageous in areas with minimal infrastructure, supporting autonomous and resilient drone operations.
Further enhancing these scenarios is the application of a multi-stage machine learning (ML) overlay, which brings capabilities like predictive analytics and adaptive routing. This advancement improves the efficiency and adaptability of the systems, significantly elevating the overall performance and user experience.”
We believe these scenarios clearly illustrate the potential and necessity of Fog Computing over NDN in dynamic and demanding environments. They demonstrate the tangible benefits and innovations that our approach can offer.
Comment 2: The proposed approach may have some scalability issues. The initial search relies on the broadcast, which will flood the network (though broadcast storm can be controlled using the existing techniques). You may consider some techniques developed for Peer2Peer Networks, such as Gnutella.
Reply.
Thank you for your constructive comments on the scalability issues inherent in our proposed approach, particularly concerning the initial broadcast search mechanism. You are correct in noting that this mechanism can lead to network flooding, a common challenge in such architectures.
In our implementation, while the initial stage indeed involves broadcasting, it is essential to recognize that this is a characteristic trait of Named Data Networking (NDN), especially during the initial search phase. To mitigate the associated overheads, we have incorporated an adaptive algorithm based on machine learning. This algorithm allows for dynamic discovery of data sources, thus progressively reducing the need for broad broadcasting as the network becomes more 'aware' of data locations.
While we acknowledge that it is challenging to eliminate the overheads of broadcasting entirely due to the inherent nature of NDN, our solution provides a balance by minimizing its impact through learning-based adaptation. Our approach significantly reduces the frequency and scope of broadcasts as the system continuously learns and adapts to the network topology and data request patterns.
Your suggestion to consider techniques from Peer 2 Peer networks like Gnutella is indeed valuable, and we plan to explore these avenues in our future work to further refine our approach.
Comment 3: Most references are old, and it needs to add some 2022 and 2023 references, e.g.
Baugh, J.; Guo, J. Enhancing Cache Robustness in Information-Centric Networks: Per-Face Popularity Approaches. Network 2023, 3, 502-521.
Liu, Z.; Jin, X.; Li, Y.; Zhang, L. NDN-Based Coded Caching Strategy for Satelite Networks. Electronics 2023, 12, 3756.
Reply.
Thank you very much for your valuable suggestion to include more recent references in our manuscript. We have now incorporated the following contemporary papers into our literature review in the “Related work” section:
- Baugh, J.; Guo, J. "Enhancing Cache Robustness in Information-Centric Networks: Per-Face Popularity Approaches." Network 2023, 3, 502-521.
- Liu, Z.; Jin, X.; Li, Y.; Zhang, L. "NDN-Based Coded Caching Strategy for Satellite Networks." Electronics 2023, 12, 3756.
We acknowledge that some of the previously cited references are indeed somewhat dated. Our initial focus was on foundational works that laid the groundwork for the field, hence the inclusion of these older yet seminal papers. However, we recognize the importance of keeping abreast of the latest advancements and have therefore updated our references to reflect recent developments in the field.
We believe these additions will significantly enrich the context and relevance of our research, offering readers a more comprehensive view of the current state of the art in the domain of Named Data Networking.
Thank you again for your insightful feedback.

Reviewer 4 Report
Comments and Suggestions for Authors
Please include detailed benchmark analysis to demonstrate the advantage of the proposed method. This is currently not clear from the manuscipt.
Author Response
Dear Reviewer:
Thank you very much for sharing your expert opinions on our work. We appreciate the time and effort taken in reviewing this submission. As a result, we are confident that our paper has benefited considerably from your constructive comments and suggestions, which were extremely helpful in improving the quality of this manuscript.
We have also made every effort to eliminate all of the indicated flaws and inconsistencies and do hope that as a result, the quality of our manuscript has improved further. The focused point-to-point answers to the review comments follow, where the feedback from the Reviewers is given in regular font, while our comments are highlighted in bold font.
Comment:
Please include detailed benchmark analysis to demonstrate the advantage of the proposed method. This is currently not clear from the manuscript.
Reply.
Thank you for your valuable feedback regarding the need for a detailed benchmark analysis. We appreciate the emphasis on the importance of demonstrating the advantages of our proposed method clearly.
In our study, we have conducted a thorough analysis for various parameters, including traffic type, caching strategies, node velocity, network size, and node density, to reflect the system's response under diverse conditions. We base our comparison on the study by Pirmagomedov et al. (2020) [1]. However, it is important to note that a direct comparison with this baseline might not be fully appropriate due to significant enhancements in our approach, such as learning-based routing and dynamic faces. These advancements lead to a fundamental shift in the operation and performance characteristics of the system, highlighting its novelty in the field.
Still we utilize the baseline performance when discussing the presented numerical results. To not replicate already published results in this work and to further clarify this to our readers we have added the following notes to the beginning of the Section 4 “Performance evaluation”:
“We note that the baseline results for the system without the proposed enhancements are provided in [X]. We will refer to them throughout this section when discussing the presented results.”
We hope this clarification addresses your concerns regarding the benchmark analysis and the unique contributions of our proposed method.
[X] Pirmagomedov, R., Srikanteswara, S., Moltchanov, D., Arrobo, G., Zhang, Y., Himayat, N., & Koucheryavy, Y. (2020, December). Augmented computing at the edge using named data networking. In 2020 IEEE Globecom Workshops (GC Wkshps (pp. 1-6). IEEE.

Round 2
Reviewer 1 Report
Comments and Suggestions for Authors
The comments are well incorporated in the revised submission.
Author Response
Dear Reviewer,
Thank you for your feedback on our manuscript. We are grateful for your positive evaluation of the relevance of our cited references and the appropriateness of our research design and methodology. We acknowledge your suggestions for improvement in the introduction, methods, results presentation, and conclusions. We will carefully revise these sections to enhance clarity and depth, ensuring our paper meets the highest standards. Your insights are invaluable to refining our work.
Reviewer 2 Report
Comments and Suggestions for Authors
line146
In wireless deployments, such as Wi-Fi, the current version of the NFD utilizes only one Face to communicate over a physical layer, see Fig.2. If the Face is configured for communication in unicast mode, then only one destination address can be specified in the remoteUri field.
NFD has supported UnicastEthernetTransport since commit 6bd6d0b50a7046fb20e7605f0aeadf35963a458d (merged on 2017-03-31), which can create a separate unicast face for each peer, on both wired and wireless Ethernet adapter.
Thus, for the same physical wireless interface, NFD can utilize zero or more unicast faces (one toward each remote MAC address), in addition to one multicast face.
line157
when a user/node supports multiple wireless interfaces simultaneously (e.g., mesh networking), enabling the targeted (unicast or multicast) communication becomes more complicated and is not supported by the default NFD implementation.
In NFD, each multicast Ethernet face is bound to a specific network interface.
As part of UnicastEthernetTransport implementation, each unicast face is part of an EthernetChannel that is bound to a specific network interface.
For both unicast and multicast communication, NFD by default supports targeted communication i.e. sending through a specific network interface.
This mechanism is supported on both wired and wireless Ethernet adapters, as NFD does not distinguish between them.
line161
Specifically, the NFD should be able to assign Faces to neighbor nodes dynamically and automatically update those changes in FIB as well as in other relevant data structures of the forwarding pipeline based on information received from the data-link layer.
This is exactly what UnicastEthernetTransport implementation is already doing.
With an EthernetChannel listening on a network interface, whenever NFD receives an NDN packet from a peer, if a unicast face does not exist, one will be created automatically.
When a peer has been inactive for a long time (configurable, default is several minutes), the unicast face is automatically closed, and any records in FIB/PIT/RIB bound to this face are deleted as well.
line318
The proposed dynamic Face management system augments the default NFD functionality with virtual Faces and their management algorithm. Multiple virtual Faces can be created for a single wireless interface (Wi-Fi NIC).
There is no need to introduce a new concept of "Virtual Faces".
It is the role of unicast faces in NFD.
line321
When a neighbor node leaves the communications range, its associated Face and corresponding prefixes reached via this Face are removed from FIB and simultaneously stored in a separate structure, named quarantine database (QDB), for a certain time. If the connection is restored within the quarantine time, the corresponding Face with associated prefixes are also restored in the FIB.
In existing NFD behavior:
* When a neighbor node leaves the communications range, NFD face system does not know about it, but the forwarding strategy will detect that Interests forwarded to this next-hop are not being satisfied and start trying other next-hops.
* The forwarding strategy remembers that this next-hop isn't working in the Measurements table, but will periodically retry this face.
* After some time (equivalent to the "quarantine time"), the face is deleted by the face system.
In your proposal:
* When a neighbor node leaves the communications range and the local node somehow knows about it, the face is deleted, and the associated FIB entries are moved into QDB.
* The forwarding strategy would not see any inaccessible next-hops.
* If the neighbor node reappears before the quarantine time has elapsed, the face is re-created and the FIB entries are re-inserted.
An overhead in your proposal is "a neighbor node leaves the communications range and the local node somehow knows about it".
For the local node to know about a neighbor node leaving the communications range quickly, you will need to periodically hear from the neighbor node.
If the neighbor node needs to periodically broadcast its presence, and the broadcast message is not part of natural Interest-Data exchanges, this would be an overhead.
In contrast, NFD's existing behavior relies on the naturally occuring Interest-Data exchanges to replace this broadcast, so that there's no additional overhead.
However, NFD's existing forwarding strategy cannot conclude "a neighbor node has left the communications range" on a per-face basis, but only operates on a per-next-hop basis, so that it could cause some forwarding delays.
It's hard to say which approach is better, but you should evaluate both approaches and compare the airtime overhead.
line340
With the dynamic Face management system, wireless devices can communicate in unicast mode, fully exploiting the benefits of advanced medium access mechanisms (such as RTS/CTS in Wi-Fi) instead of using random access channel variations for transmitting user data. Moreover, in unicast mode, wireless technologies support error-control methods, such as automatic repeat request (ARQ), further improving communication reliability. Eventually, the dynamic Face management system promises a better user experience for customers in wireless mesh networking or other multi-connectivity scenarios.
Even if you can justify and evaluate the benefit of having QDB, the "dynamic Face management system" is not noval, as it's part of UnicastEthernetTransport implementation.
Hence, the resulting benefits of having unicast communication is not a contribution of your design.
line385
To enable fog computing in dynamic wireless meshes, we developed a new forwarding strategy named Windowed Strategy.
I do not see any difference between Windowed Strategy and self-learning strategy (with enhancements in Teng Liang's papers, not yet implemented in mainline NFD).
Author Response
Dear Reviewer,
Thank you for your meticulous review and insightful observations on our manuscript. Below are our responses to each of your comments.
Comment 1:
line146
“In wireless deployments, such as Wi-Fi, the current version of the NFD utilizes only one Face to communicate over a physical layer, see Fig.2. If the Face is configured for communication in unicast mode, then only one destination address can be specified in the remoteUri field."
NFD has supported UnicastEthernetTransport commit 6bd6d0b50a7046fb20e7605f0aeadf35963a458d (merged on 2017-03-31), which can create a separate unicast face for each peer, on both wired and wireless Ethernet adapter. Thus, for the same physical wireless interface, NFD can utilize zero or more unicast faces (one toward each remote MAC address), in addition to one multicast face.
line157
“when a user/node supports multiple wireless interfaces simultaneously (e.g., mesh networking), enabling the targeted (unicast or multicast) communication becomes more complicated and is not supported by the default NFD implementation.”
In NFD, each multicast Ethernet face is bound to a specific network interface. As part of UnicastEthernetTransport implementation, each unicast face is part of an EthernetChannel that is bound to a specific network interface. For both unicast and multicast communication, NFD by default supports targeted communication i.e. sending through a specific network interface. This mechanism is supported on both wired and wireless Ethernet adapters, as NFD does not distinguish between them.
Response
Thank you for your insightful observation regarding the Named Data Networking Forwarding Daemon (NFD) and its capabilities in managing multiple unicast faces. We acknowledge the significance of UnicastEthernetTransport in enabling NFD to handle several unicast faces for each peer on both wired and wireless Ethernet adapters.
Unfortunately, at the moment of performing the research reflected in the paper, we were unaware of the solutions you point out to. At the revision phase, we explicitly acknowledged the work has been done so far.
We have added the following description of the current implementation
“The Named Data Networking Forwarding Daemon (NFD) is a crucial component in the architecture of Named Data Networking (NDN). The integration of UnicastEthernetTransport has significantly enhanced NFD's capabilities in managing faces. UnicastEthernetTransport enables NFD to create a separate unicast face for each peer on both wired and wireless Ethernet adapters. UnicastEthernetTransport in NFD is designed to handle unicast communications over Ethernet. It supports the creation of individual unicast faces for each peer, which is crucial for targeted communication in NDN. This transport mechanism allows for efficient management of network interfaces, binding each unicast face to a specific network interface. This is particularly beneficial in environments with multiple interfaces, such as mesh networks. It supports both unicast and multicast modes of communication, making it versatile for various networking scenarios. Unicast mode is particularly useful for direct, point-to-point communication, while multicast mode facilitates the distribution of data to multiple recipients. One of the key features of UnicastEthernetTransport is its adaptability to both wired and wireless Ethernet adapters, treating them uniformly. This enhances the flexibility of NFD in different network setups. The integration of UnicastEthernetTransport significantly extends the capabilities of NFD, particularly in handling complex network structures and traffic patterns more efficiently. UnicastEthernetTransport plays a vital role in enhancing the capabilities of NFD within NDN. Its ability to manage multiple faces dynamically and support both unicast and multicast communications across wired and wireless interfaces positions NFD as a versatile and powerful tool in the network data management.”
To attract readers attention to these facts we highlighted in the introduction that we refine the solutions proposed so far., e.g.
“This paper fills the identified gaps and refines the solutions developed so far by proposing two mechanisms enabling efficient NDN-based Fog computing services in wireless mesh systems…”
Comment 2:
line161
“Specifically, the NFD should be able to assign Faces to neighbor nodes dynamically and automatically update those changes in FIB as well as in other relevant data structures of the forwarding pipeline based on information received from the data-link layer."
This is exactly what UnicastEthernetTransport implementation is already doing.
With an EthernetChannel listening on a network interface, whenever NFD receives an NDN packet from a peer, if a unicast face does not exist, one will be created automatically. When a peer has been inactive for a long time (configurable, default is several minutes), the unicast face is automatically closed, and any records in FIB/PIT/RIB bound to this face are deleted as well.
Response
We appreciate your clarification regarding the dynamic assignment of Faces in the Named Data Networking Forwarding Daemon (NFD), particularly with its UnicastEthernetTransport implementation. This insight has allowed us to re-evaluate our understanding of the NFD's functionalities, especially in the context of dynamic face management.
Our manuscript, in its current form, does briefly touch upon the dynamic assignment and updating of Faces in NFD. However, based on your observation, we recognize the need for a more explicit and detailed exposition of this mechanism. To this end, we will slightly adjust our discussion to better articulate how the NFD, through the UnicastEthernetTransport, efficiently manages dynamic face assignment and updates the FIB and other relevant data structures. This involves the automatic creation of a unicast face when an NDN packet is received from a peer and the deletion of the face and related records in FIB/PIT/RIB after a period of inactivity.
We aim to ensure that this nuanced elaboration on NFD's face management capabilities is seamlessly integrated into our existing discussion, providing a clearer and more comprehensive understanding of these functionalities without necessitating extensive revisions.
Your feedback is invaluable in guiding these refinements, and we are committed to enhancing the clarity and accuracy of our manuscript in light of your insights.
Comment 3:
line318
“The proposed dynamic Face management system augments the default NFD functionality with virtual Faces and their management algorithm. Multiple virtual Faces can be created for a single wireless interface (Wi-Fi NIC).”
There is no need to introduce a new concept of "Virtual Faces".
It is the role of unicast faces in NFD.
line321
“When a neighbor node leaves the communications range, its associated Face and corresponding prefixes reached via this Face are removed from FIB and simultaneously stored in a separate structure, named quarantine database (QDB), for a certain time. If the connection is restored within the quarantine time, the corresponding Face with associated prefixes are also restored in the FIB.”
In existing NFD behavior:
* When a neighbor node leaves the communications range, NFD face system does not know about it, but the forwarding strategy will detect that Interests forwarded to this next-hop are not being satisfied and start trying other next-hops.
* The forwarding strategy remembers that this next-hop isn't working in the Measurements table, but will periodically retry this face.
* After some time (equivalent to the "quarantine time"), the face is deleted by the face system.
In your proposal:
* When a neighbor node leaves the communications range and the local node somehow knows about it, the face is deleted, and the associated FIB entries are moved into QDB.
* The forwarding strategy would not see any inaccessible next-hops.
* If the neighbor node reappears before the quarantine time has elapsed, the face is re-created and the FIB entries are re-inserted.
An overhead in your proposal is "a neighbor node leaves the communications range and the local node somehow knows about it". For the local node to know about a neighbor node leaving the communications range quickly, you will need to periodically hear from the neighbor node. If the neighbor node needs to periodically broadcast its presence, and the broadcast message is not part of natural Interest-Data exchanges, this would be an overhead.
In contrast, NFD's existing behavior relies on the naturally occurring Interest-Data exchanges to replace this broadcast, so that there's no additional overhead. However, NFD's existing forwarding strategy cannot conclude "a neighbor node has left the communications range" on a per-face basis, but only operates on a per-next-hop basis, so that it could cause some forwarding delays. It's hard to say which approach is better, but you should evaluate both approaches and compare the airtime overhead.
Response
Thank you for your additional clarifications. Comparing the overheads for the current implementation and the one we have proposed is indeed a very interesting idea. Unfortunately, the journal does not provide sufficient time to provide any meaningful comparison campaign. However, to highlight that this task is really interesting for a further look we provided a short note in the conclusions part of the paper as follows:
“Finally, we would like to mention that the proposed and the current Face implementations introduce different types of overheads in dynamic network conditions. The comparison of these overheads is a part of our future work.”
Comment 4:
line 340
“With the dynamic Face management system, wireless devices can communicate in unicast mode, fully exploiting the benefits of advanced medium access mechanisms (such as RTS/CTS in Wi-Fi) instead of using random access channel variations for transmitting user data. Moreover, in unicast mode, wireless technologies support error-control methods, such as automatic repeat request (ARQ), further improving communication reliability. Eventually, the dynamic Face management system promises a better user experience for customers in wireless mesh networking or other multi-connectivity scenarios.”
Even if you can justify and evaluate the benefit of having QDB, the "dynamic Face management system" is not noval, as it's part of UnicastEthernetTransport implementation. Hence, the resulting benefits of having unicast communication is not a contribution of your design.
Response
Thank you, now we understand that we missed one of the critical functionalities when doing a related work review. We also agree that the Face management system has already been implemented prior to our work.
To not confuse the readers, we have removed the dynamic face management system from the list of our contributions leaving just the performance evaluation of it jointly with the proposed learning strategy, i.e., the third contribution now reads as:
“performance evaluation of the adaptive learning strategy and dynamic Face management system in dynamic network conditions showing that the suggested enhancements in NDN system design efficiently support fog computing in multi-hop wireless mesh systems.”
Comment 5
line385
“To enable fog computing in dynamic wireless meshes, we developed a new forwarding strategy named Windowed Strategy.”
I do not see any difference between Windowed Strategy and self-learning strategy (with enhancements in Teng Liang's papers, not yet implemented in mainline NFD).
We guess you refer to the following paper:
“Liang, T., Pan, J., Rahman, M. A., Shi, J., Pesavento, D., Afanasyev, A., & Zhang, B. (2020, June). Enabling named data networking forwarder to work out-of-the-box at edge networks. In 2020 IEEE International Conference on Communications Workshops (ICC Workshops) (pp. 1-6). IEEE.”
Indeed, the proposed Windowed Strategy is conceptually similar to the self-learning strategy in [1]. However, the proposed strategy makes an attempt to find and maintain all (or at least a set of) possible routes by not only forwarding an interest through the already known route, but also by probing other faces.. The probe is limited by specific exponential timers attached to each face in order to limit the network flooding - the less successful data receptions you get through a given face, the less often you will probe it, while successful reception will increase the lifetime of the route.
At the same time, we did not cover any security questions and related route discovery issues as it was outside the work scope and involved more than just altering a forwarding strategy.
To reflect this in the paper, we have added the following to the Section 3.2 “Adaptive Forwarding Strategy”
“We specifically note that the proposed Windowed Strategy is conceptually similar to the self-learning strategy in [1]. However, the proposed strategy makes an attempt to find and maintain all (or at least a set of) possible routes by not only forwarding an interest through the already known route, but also by probing other faces.. The probe is limited by specific exponential timers attached to each face in order to limit the network flooding - the less successful data receptions you get through a given face, the less often you will probe it, while successful reception will increase the lifetime of the route.”
[1] Liang, T., Pan, J., Rahman, M. A., Shi, J., Pesavento, D., Afanasyev, A., & Zhang, B. (2020, June). Enabling named data networking forwarder to work out-of-the-box at edge networks. In 2020 IEEE International Conference on Communications Workshops (ICC Workshops) (pp. 1-6). IEEE.
Reviewer 4 Report
Comments and Suggestions for Authors
could not see sufficient benchmark proposed and conducted
Comments on the Quality of English Languagegenerally fine. minor edits recommended.
Author Response
Dear Reviewer 1,
We sincerely appreciate your time and effort in reviewing our manuscript. Here are our responses to your comments:
Comment 1 on the Quality of English Language: generally fine. minor edits recommended.
We have made every effort to eliminate all of the indicated flaws and inconsistencies and do hope that as a result, the quality of our manuscript has improved further.
Comment 2: could not see sufficient benchmark proposed and conducted.
Thank you for highlighting the need for clarity in our benchmarking approach. We acknowledge your concerns and have revisited our analysis to ensure technical precision and scientific rigor.
In our study, we employed a comprehensive benchmarking methodology, focusing on a range of parameters that include traffic type, caching strategies, node velocity, network size, and node density. This approach was designed to assess the system's response under diverse operational conditions, ensuring a thorough understanding of its performance capabilities.
We based our benchmark comparison on the foundational study by Pirmagomedov et al. (2020) [1]. It is essential to note, however, that our study introduces significant methodological advancements, such as the integration of learning-based routing and dynamic face management. These changes contribute to a shift in the operation and performance characteristics of the system, thus extending beyond the baseline model's scope.
To ensure transparency and provide context to our readers, we have included a specific statement in Section 4, "Performance Evaluation," of our manuscript:
"We acknowledge that the baseline results for the system without our proposed enhancements are detailed in [1]. Our discussion in this section references these baseline metrics to underscore the improvements achieved through our advanced routing and face management strategies."
This approach is intended to demonstrate the incremental advancements of our system compared to the baseline, while not replicating previously published results. We believe that this complies with the baseline study effectively and highlights the contributions and the enhanced capabilities of our proposed method.
We trust that this clarification addresses your concerns and accurately reflects the scope and depth of our benchmark analysis.
[1] Pirmagomedov, R., Srikanteswara, S., Moltchanov, D., Arrobo, G., Zhang, Y., Himayat, N., & Koucheryavy, Y. (2020, December). Augmented computing at the edge using named data networking. In 2020 IEEE Globecom Workshops (GC Wkshps (pp. 1-6). IEEE.
Round 3
Reviewer 2 Report
Comments and Suggestions for Authors
Follow-up to Response 1
Unfortunately, at the moment of performing the research reflected in the paper, we were unaware of the solutions you point out to. At the revision phase, we explicitly acknowledged the work has been done so far.
You need to delete from the paper those inaccurate statements regarding features not supported by NFD that are actually supported.
Follow-up to Response 2 and Response 4
We aim to ensure that this nuanced elaboration on NFD's face management capabilities is seamlessly integrated into our existing discussion, providing a clearer and more comprehensive understanding of these functionalities without necessitating extensive revisions.
To not confuse the readers, we have removed the dynamic face management system from the list of our contributions leaving just the performance evaluation of it jointly with the proposed learning strategy
You need to delete from the paper the improper method of "virtual faces". This should be replaced with unicast Ethernet faces in NFD.
Follow-up to Response 3
Comparing the overheads for the current implementation and the one we have proposed is indeed a very interesting idea. Unfortunately, the journal does not provide sufficient time to provide any meaningful comparison campaign.
You need to take more time to complete these important evaluations.
Follow-up to Response 5
Indeed, the proposed Windowed Strategy is conceptually similar to the self-learning strategy. However, the proposed strategy makes an attempt to find and maintain all (or at least a set of) possible routes by not only forwarding an interest through the already known route, but also by probing other faces.
The probing mechanism isn't new. See NDN Technical Report NDN-0042, An Experimental Investigation of Hyperbolic Routing with a Smart Forwarding Plane in NDN, https://named-data.net/publications/techreports/ndn-0042-1-asf/ . The implementation is ASF strategy, part of mainline NFD codebase.
Overall
The ONLY contribution of this paper is combing the probing mechanism (of ASF strategy) with the self-learning strategy.
All others are already superseded by state-of-art. You need to delete these parts and re-design the solution.
Author Response
Dear reviewer,
Thank you very much for your valuable feedback. Below are our responses to each of your comments.
Follow-up to Response 1
Unfortunately, at the moment of performing the research reflected in the paper, we were unaware of the solutions you point out to. At the revision phase, we explicitly acknowledged the work has been done so far.
You need to delete from the paper those inaccurate statements regarding features not supported by NFD that are actually supported.
Thank you for highlighting the inaccuracies regarding the features of NFD in our manuscript. We acknowledge the oversight in our initial understanding of the NFD's capabilities, particularly those related to UnicastEthernetTransport.
We have taken your feedback into consideration and have revised the manuscript to accurately reflect the current functionalities supported by NFD. The necessary corrections have been made to ensure that our paper presents an accurate representation of NFD's features, specifically the handling of unicast and multicast faces in wireless and wired network deployments.
Follow-up to Response 2 and Response 4
We aim to ensure that this nuanced elaboration on NFD's face management capabilities is seamlessly integrated into our existing discussion, providing a clearer and more comprehensive understanding of these functionalities without necessitating extensive revisions.
To not confuse the readers, we have removed the dynamic face management system from the list of our contributions leaving just the performance evaluation of it jointly with the proposed learning strategy
You need to delete from the paper the improper method of "virtual faces". This should be replaced with unicast Ethernet faces in NFD.
We agree with your assessment and acknowledge that the concept of "virtual faces" as initially presented in our manuscript does not accurately represent the functionalities provided by NFD's UnicastEthernetTransport. To rectify this, we have revised the manuscript to eliminate references to "virtual faces." Instead, we now correctly attribute the functionalities to unicast Ethernet faces as implemented in NFD.
Follow-up to Response 5
Indeed, the proposed Windowed Strategy is conceptually similar to the self-learning strategy. However, the proposed strategy makes an attempt to find and maintain all (or at least a set of) possible routes by not only forwarding an interest through the already known route, but also by probing other faces.
The probing mechanism isn't new. See NDN Technical Report NDN-0042, An Experimental Investigation of Hyperbolic Routing with a Smart Forwarding Plane in NDN, https://named-data.net/publications/techreports/ndn-0042-1-asf/ . The implementation is ASF strategy, part of mainline NFD codebase.
Thank you for your detailed comments and links to the corresponding materials. In response, we have removed the sections of our manuscript that discuss the "Windowed Strategy" and other elements that are already addressed by the current state-of-the-art, and mentioned the contribution of combining the self-learning strategy with the probing mechanism from the ASF strategy.
Follow-up to Response 3
Comparing the overheads for the current implementation and the one we have proposed is indeed a very interesting idea. Unfortunately, the journal does not provide sufficient time to provide any meaningful comparison campaign.
You need to take more time to complete these important evaluations.
Thank you once again for your insightful suggestion regarding the comparison of overheads between the current implementation and our proposed system. We recognize the value such a comparison would bring to our research, offering a more comprehensive understanding of the performance and efficiency implications of our proposed approach.
However, as we have noted before, the time constraints imposed by the journal's publication schedule present a significant challenge in conducting a thorough and meaningful comparison campaign. These evaluations require meticulous planning, execution, and analysis to ensure accuracy and relevance, all of which demand a considerable amount of time.
Overall
The ONLY contribution of this paper is combing the probing mechanism (of ASF strategy) with the self-learning strategy.
All others are already superseded by state-of-art. You need to delete these parts and re-design the solution.
We deeply appreciate the insights and recommendations provided. In light of the existing time constraints, we propose the following course of action:
-
Submission of Current Research: We will proceed with the submission of our current research findings with suggested revisions, which we believe still provide a contribution to the field as they stand.
-
Future Work: We commit to undertaking the suggested comparison as part of our ongoing research. This will allow us to devote the necessary time and resources to ensure a robust and comprehensive analysis.
-
Subsequent Publication: Upon completion of this additional comparison campaign, we plan to prepare a follow-up study detailing our findings. This study would be submitted as a separate paper, allowing us to share these important insights with the community in a timely manner.
In conclusion, we extend our sincere gratitude to the reviewer for thorough analysis and constructive feedback on our paper. The depth of expertise and the detail in the suggestions have been invaluable in guiding the essential revisions of our study. We are immensely thankful for the opportunity to enhance our work through this collaborative and rigorous peer-review process.
Reviewer 4 Report
Comments and Suggestions for Authors
For benchmark analysis, a wider selection of other approaches should be considered. In the meanwhile, computer codes/data should be provided for replication. Finally, benchmark analysis should include testing for statistical significance of differences in different models.
Author Response
Dear Reviewer,
Thank you for your valuable comments and suggestions. In response to your review:
1. Benchmark Analysis Scope: Our numerical results explore the characteristics of the system as a function of various system parameters, including but not limited to those outlined in Pirmagomedov et al. [1]. This approach allows us to address a broader spectrum of scenarios, especially those involving wireless and mobile networks, which are central to our study. It is important to note that many existing works focus on different contexts, often considering stationary network environments. Hence, a direct comparison with static network models would not correctly reflect the dynamic nature and challenges specific to our study's focus on wireless and mobile networks.
2. Statistical Significance and Data Presentation: Regarding the statistical significance of our results, we ensured the collected data volume was sufficient to achieve a confidence interval of no more than 3% from the point estimates. This level of precision in our data allowed us to confidently present only point estimates in our work, as the narrow confidence intervals consistently reinforced the reliability of these estimates. By focusing on point estimates with such high confidence, we aim to provide clear and concise insights into the performance and behavior of the system under various conditions.
We hope these clarifications address your concerns regarding our benchmark analysis approach and the representation of our results.
Reference:
[1] Pirmagomedov, R., Srikanteswara, S., Moltchanov, D., Arrobo, G., Zhang, Y., Himayat, N., & Koucheryavy, Y. (2020, December). Augmented computing at the edge using named data networking. In 2020 IEEE Globecom Workshops (GC Wkshps) (pp. 1-6). IEEE.